# Reward Learning through Ranking Mean Squared Error

## Abstract

Reward design remains a significant bottleneck in applying reinforcement learning (RL) to real-world problems. A popular alternative is reward learning, where reward functions are inferred from human feedback rather than manually specified. Recent work has proposed learning reward functions from human feedback in the form of ratings, rather than traditional binary preferences, enabling richer and potentially less cognitively demanding supervision. Building on this paradigm, we introduce a new rating-based RL method, Ranked Return Regression for RL (R4). At its core, R4 employs a novel ranking mean squared error (rMSE) loss, which treats teacher-provided ratings as ordinal targets. Our approach learns from a dataset of trajectory–rating pairs, where each trajectory is labeled with a discrete rating (e.g., "bad," "neutral," "good"). At each training step, we sample a set of trajectories, predict their returns, and rank them using a differentiable sorting operator (soft ranks). We then optimize a mean squared error loss between the resulting soft ranks and the teacher's ratings. Unlike prior rating-based approaches, R4 offers formal guarantees: its solution set is provably minimal and complete under mild assumptions. Empirically, using simulated human feedback, we demonstrate that R4 consistently matches or outperforms existing rating and preference-based RL methods on robotic locomotion benchmarks from OpenAI Gym and the DeepMind Control Suite, while requiring significantly less feedback.

## 1 Introduction

Deep reinforcement learning (RL) has achieved remarkable success in games, where well-defined reward functions are available (Mnih et al., 2015; Schrittwieser et al., 2019). In contrast, real-world environments often lack clean specifications, making reward design a significant bottleneck to deploying RL in complex, practical applications (Knox et al., 2023; Knox & MacGlashan, 2024). In practice, reward design is often an informal trial-and-error process where RL practitioners iteratively adjust a reward function until the RL agent exhibits acceptable behavior (Booth et al., 2023). This procedure can be error-prone, resulting in reward misspecification where practitioners inadvertently define a reward function that does not align with the true task objective. This can result in the agent learning undesirable or unintended behaviors (Skalse et al., 2022; Pan et al., 2022). Notably, these challenges have been observed even in tabular domains, illustrating that reward design remains a core challenge, even in simple settings (Muslimani et al., 2025).

A popular alternative to manual reward engineering is *reward learning*, where reward functions are inferred from human feedback rather than explicitly designed. This feedback can take various forms, including scalar evaluations (Knox & Stone, 2009; MacGlashan et al., 2017), demonstrations (Taylor, 2018; Arora & Doshi, 2021), or pairwise preferences over agent behaviors (Christiano et al., 2017b). Of these approaches, preference-based RL has gained particular traction due to its low human effort and its role in large language models (OpenAI et al., 2024).

Despite its success, learning from binary preferences can be limiting. For one, each binary comparison conveys only a single bit of information, making reward learning sample inefficient in terms of required preference labels. As a result, more human time may be required compared to collecting multi-class ratings, increasing the overall feedback burden. Moreover, such feedback is inherently relative. It indicates which behavior is preferred, but not by how much, nor whether either option is

good in an absolute sense. For example, if both behaviors under comparison are poor, a human can at best indicate that they are equally preferable, but cannot express that both are low-quality overall.

Recent work has introduced a new paradigm known as rating-based reinforcement learning (RbRL) (White et al., 2024), in which humans provide discrete, multi-class ratings rather than binary preferences to guide reward learning. Instead of comparing two behaviors and selecting the preferred one, the human observes a single behavior at a time and assigns it a rating from a fixed scale. This shift enables the collection of richer feedback, as ratings can capture both relative and absolute assessments of trajectory quality. Moreover, compared to preference-based feedback, user studies have found that participants perceive rating-based feedback as less cognitively demanding, and report feeling more successful when completing tasks using ratings (White et al., 2024).

The advantages of RbRL motivate the development of more efficient algorithms for learning from ratings. To this end, we propose a new rating-based RL method: Ranked Return Regression for RL (R4). It learns reward functions from trajectories labeled with ordinal ratings using a novel ranking mean squared error (rMSE) loss. At each training step, we sample one trajectory per rating class, compute their predicted returns under the reward model, and rank them using a differentiable sorting operator (Blondel et al., 2020) (i.e., soft ranks). We then minimize a mean squared error loss between the resulting soft ranks and the teacher's ratings.

Our contributions are as follows:

1. We propose Ranked Return Regression for RL, a rating-based RL algorithm that leverages a novel ranking mean squared error loss to train reward functions from trajectories labeled with ordinal ratings.

2. We establish that rMSE is the first rating-based RL objective with provable minimality and completeness guarantees under mild assumptions.

3. We empirically validate R4 in both offline and online feedback settings, demonstrating it can outperform RbRL and other preference-based RL algorithms in several robotic locomotion tasks from OpenAI Gym (Brockman et al., 2016) and the DeepMind Control (DMC) Suite (Tassa et al., 2018).

Taken together, these contributions emphasize rating-based RL methods that are both theoretically grounded and effective in leveraging teacher feedback.

## 2 RELATED WORK

Reward learning is a broad field in which reward functions are inferred from various forms of human feedback, including demonstrations, preferences, scalar evaluations, ratings, or combinations thereof. One common approach is inverse reinforcement learning (IRL), which learns a reward function such that the resulting policy produces behaviors similar to those in the provided demonstrations (Ng & Russell, 2000). Despite recent advances showing that reward functions can be recovered from suboptimal demonstrations (Shiarlis et al., 2016; Brown et al., 2019), it is still argued that providing demonstrations can be time-consuming and difficult (Akgun et al., 2012; Lee et al., 2021).

As an alternative to demonstrations, other approaches rely on preference-based feedback (e.g., preference-based RL). In this setting, human users typically provide binary preferences over pairs of agent behaviors (Christiano et al., 2017a; Lee et al., 2021). This form of supervision has gained recent popularity, as it is often considered more intuitive and less demanding than providing full demonstrations. However, binary preferences can be limited in the richness of information they convey. To address this, recent work has explored scaled preferences, where users indicate not just which behavior they prefer, but also the strength of that preference (e.g., on a scale from "strongly prefer A" to "strongly prefer B") (Wilde et al., 2021). These graded comparisons have been shown to outperform strict binary preferences, offering more informative supervision for reward learning.

Similarly, scalar feedback methods provide rich signals by allowing humans to rate behaviors directly. For example, the TAMER framework allowed humans to provide binary signals indicating whether a behavior was judged to be optimal (Knox & Stone, 2009). Later work introduced reward sketching, where, for a given behavior, humans continuously provide a scalar signal indicating the agent's progress toward a goal (Cabi et al., 2019). Recent work on reward modeling for LLMs has

also moved beyond binary preferences to use ordinal feedback (Liu et al., 2025a;b). Most similar to our approach is rating-based RL (White et al., 2024). It handles multi-class discrete ratings by using a novel cross-entropy loss formulation. In this setup, humans assign class labels such as "good," "okay," or "bad" to trajectories, and these labels are then used to train a reward function.

In addition to the type of feedback, reward learning can also be categorized by how feedback is collected and used. In the offline setting, reward models are trained on static datasets of labeled trajectories (i.e., labeled with preferences, ratings, etc) and the learned reward is then used to train an RL agent. In the online setting, the agent interacts with the environment and receives feedback in real time; trajectories generated by the agent are periodically labeled, and the reward model is updated continuously as the agent learns its policy.

## 3 BACKGROUND

**Markov Decision Processes Without Rewards** denoted MDP\R, is an MDP where feedback is provided in the form of teacher ratings (Abbeel & Ng, 2004; White et al., 2024). Formally, MDP\R is defined as: $\mathcal{M} = (\mathcal{S}, \mathcal{A}, T, \rho, \gamma, n, \mathcal{D})$, where $\mathcal{S}$ and $\mathcal{A}$ are the state and action spaces, $T : \mathcal{S} \times \mathcal{A} \times \mathcal{S} \to [0, 1]$ is the transition probability function, $\rho$ is the initial state distribution, and $\gamma \in [0, 1)$ is the discount factor. Note that $\gamma$ is fixed (not learned), as is standard in reward learning. Specifically, a teacher watches each trajectory $\tau_i$, where $\tau_i = (s_1^i, a_1^i, \ldots, s_T^i, a_T^i)$, and assigns it a rating $c_i \in 0, 1, \ldots, n - 1$, where $c_i$ indicates the perceived quality of the trajectory. A rating of 0 represents the lowest quality, while $n - 1$ represents the highest. We define the function $c(\tau_i)$ as the function that maps the trajectory $\tau_i$ to the corresponding rating class. Note that the rating classes can also be assigned descriptive labels to aid interpretation. For instance, with $n = 3$ rating classes, the labels might be: 0 — "bad," 1 — "neutral," and 2 — "good". This process is repeated for all trajectories, and the resulting data is grouped by rating class. Specifically, for each rating class $k \in \{0, 1, \ldots, n - 1\}$, we define a subset $\mathcal{D}_k = \{\tau_i \mid c(\tau_i) = k\}$ containing all trajectories assigned to rating class $k$. The full dataset is then $\mathcal{D} = \bigcup_{k=0}^{n-1} \mathcal{D}_k$.

Compared to the standard RL setting, the key difference is that the MDP includes a reward function $r : \mathcal{S} \times \mathcal{A} \to \mathbb{R}$, which provides a numerical reward for each state-action pair. This replaces the teacher ratings component (e.g., $n$, $\mathcal{D}$) used in our setting. The objective is then to find an optimal policy $\pi^*$ that maximizes the expected discounted return, $G_r$, defined as: $\sum_t \gamma^t r(s_t, a_t)$. In contrast, the MDP\R setting lacks an engineered reward function, and thus the goal becomes two-fold: (1) to learn a parameterized reward function $\hat{r}_\theta$ from teacher-provided ratings; and (2) to learn a policy that maximizes the expected discounted return with respect to $\hat{r}_\theta$, such that the resulting policy produces behaviors that statisfy the teacher's ratings. [1]

**Differentiable (Soft) Ranking** refers to a class of algorithms designed to make the sorting and ranking process differentiable (Grover et al., 2019; Blondel et al., 2020; Petersen et al., 2022). In R4, we use the algorithm proposed by Blondel et al. (2020), which assigns continuous ranks to a set of values. Unlike hard sorting, soft ranking is smooth and differentiable, allowing gradients to propagate through the ranking operation during optimization. For example, given the values $[3.2, 1.0, 4.5]$, the algorithm produces approximate soft ranks $\hat{R} = [1.5, 3.0, 1.0]$. By contrast, the true hard ranks are $R = [1, 0, 2]$, with the highest value ranked 2 and the lowest ranked 0.

**Rating Based Reinforcement Learning** is a form of reward learning introduced by White et al. (2024), where reward models are trained from discrete ratings via supervised learning, rather than from preferences. In particular, they introduced a cross-entropy–style loss function defined as:

$$\mathcal{L}_{\text{RbRL}} = \mathbb{E}_{\tau_i} \left( - \sum_{i=0}^{N-1} \mu_i \log(Q(\tau_i)) \right),$$

where $\mu_i$ is the indicator function for the assigned label (i.e., $\mu_i = 1$ when the trajectory $\tau_i$ is assigned the label class $i$ in the dataset, and 0 otherwise). Furthermore, the function $Q$ is defined as:

---

[1]Note that reward evaluation is not well defined in the literature, see (Muslimani et al., 2025).

$$Q(\tau_i) = \frac{e^{-k(\hat{G}_i - B_i)(\hat{G}_i - B_{i+1})}}{\sum_j e^{-k(\hat{G}_j - B_j)(\hat{G}_j - B_{j+1})}}$$

Here, $\{B_i\}_{i=0}^{N-1}$ are class decision boundaries and $k$ is a hyperparameter. For convenience, we write $\hat{G}_i$ instead of $\hat{G}_\theta(\tau_i)$, where $\hat{G}_i \in [0, 1]$ denotes the normalized predicted return under the learned reward model $\hat{r}_\theta$. As noted in White et al. (2024) and reproduced in section A.1, $\mathcal{L}_{\text{RbRL}}$ encourages the predicted returns of all trajectories within a class to concentrate around the midpoint $\frac{B_i + B_{i+1}}{2}$.

## 4 RANKED RETURN REGRESSION FOR RL – R4

Given a set of rating classes $c_0, \ldots, c_{n-1}$, where $i < j$ implies that trajectories in class $c_i$ are rated lower than those in class $c_j$, we assume that for any $\tau_a \in \mathcal{D}_i$ and $\tau_b \in \mathcal{D}_j$, the (unobserved) return under the the teacher's implicit reward function, $r^*$, satisfies $G^*(\tau_a) < G^*(\tau_b)$. Here, $G^*(\tau) = \sum_t \gamma^t r^*(s_t, a_t)$ denotes the discounted return of trajectory $\tau$ with respect to the teacher's reward function. While ratings assign trajectories independently to discrete classes, we can construct a ranking by ordering trajectories according to their assigned classes. Consequently, by sampling one trajectory from each class, we obtain a perfectly ordered ranking over $n$ trajectories (where $n$ is the number of rating classes).

We leverage this observation to define a novel ranking mean squared error (rMSE) objective over a set of trajectories, which serves as a supervised learning loss for training a reward function from trajectory ratings. We can then use this learned reward function, in place of an engineered reward, to train an RL policy to maximize the expected return, $\hat{G}_\theta$ denoted as $\mathbb{E}\left[\sum_t \gamma^t \hat{r}_\theta(s_t, a_t)\right]$. We refer to this rMSE-based training pipeline as Ranked Return Regression for RL. We demonstrate that this algorithm is flexible, applying it in both offline and online feedback settings.

### 4.1 REWARD LEARNING WITH THE RANKING MEAN SQUARED ERROR OBJECTIVE

To learn the reward function $\hat{r}_\theta$ from the dataset $\mathcal{D}$, we sample one trajectory $\tau_i$ from each class dataset $\mathcal{D}_k$. The return for each sampled trajectory is estimated as:

$$\hat{G}_\theta(\tau_i) = \sum_t \gamma^t \hat{r}_\theta(s_t, a_t)$$

We then rank these returns using a differentiable sorting algorithm (Blondel et al., 2020), yielding a soft rank $\hat{R}_\theta(\tau_i)$ for each $\hat{G}_\theta(\tau_i)$. The rMSE loss is computed as the mean squared error between the soft rank and the rating class provided by a teacher:

$$\mathcal{L}_{\text{rMSE}} = \frac{1}{n} \left[ \sum_{i=0}^{n-1} \left( \hat{R}_\theta(\tau_i) - c(\tau_i) \right)^2 \right] \tag{1}$$

For example, suppose the rating classes for the sampled trajectories are $c = [2, 3, 1]$, and the predicted soft ranks of returns are $\hat{R}_\theta = [1.0, 3.0, 2.0]$, We compute the rMSE loss as:

$$\mathcal{L}_{\text{rMSE}} = \frac{1}{3} \left[ (1.0 - 2.0)^2 + (3.0 - 3.0)^2 + (2.0 - 1.0)^2 \right]$$

In this example, the predicted ranks for the first and third trajectories deviate from the corresponding rating. Since the soft ranks are differentiable with respect to the reward parameters $\theta$, minimizing this loss allows the model to adjust $\hat{r}_\theta$ to better align with the ratings provided by a teacher. See Figure 1 for an overview of the R4 training process.

There are multiple advantages to using the rMSE objective over the RbRL objective:

1. **Eliminating hyperparameters:** The RbRL objective requires specifying rating class boundaries $B$. A trajectory is assigned to class $k$ only if its return lies between $B_k$ and $B_{k+1}$. Our approach does not require such explicitly defined boundaries.

2. **Preserving within-class diversity:** The RbRL objective encourages all trajectories in a class to have predicted returns close to the midpoint $\frac{B_k+B_{k+1}}{2}$, thereby ignoring intra-class diversity. In contrast, the rMSE objective does not enforce such a constraint, allowing greater flexibility in modeling returns within the same class. In Proposition 1, we show that enforcing such constraints on intra-class variability can be detrimental.

3. **Dynamic rating classes:** The rMSE objective allows the number and structure of rating classes to change dynamically during training. This flexibility means raters are not restricted to a fixed rating scheme; they can introduce new classes if they want finer distinctions or merge existing ones when coarser ratings are more natural. In contrast, extending the RbRL objective to dynamic classes is non-trivial, as its performance can degrade when the number of bins deviates from the optimal range (White et al., 2024).[2]

## 4.2 Design Choices for Online Feedback Setting

For the online feedback experiments, we implement several strategies to use teacher feedback more effectively. First, we apply a dynamic feedback schedule that collects feedback more frequently at the start of training and gradually reduces the feedback frequency as training progresses. Next, to determine which trajectory segments to sample, we maintain a dataset of the most recent 50 trajectories and apply a stratified sampling approach. We select a fraction from high-predicted-return trajectories and the remainder from lower-predicted-return ones. For each selected trajectory, we extract a sub-trajectory of fixed length, choosing either a random segment or the segment with the highest predicted return with equal probability. This heuristic aims to balance exploration of diverse trajectories with attention to promising ones. Further details about the preference collection schedule and the sampling strategy are provided in Appendix C.3. We test the impact of these strategies on learning progress in Section 6.3.

Moreover, in R4, we use dynamic rating classes to better accommodate how teachers might provide feedback. Early in training, when the agent produces mostly low-quality behavior, we use finer-grained bins to distinguish poor trajectories, allowing teachers to provide more informative feedback. As higher-quality trajectories emerge, these bins are merged into a single "low quality" class, and when a trajectory's return falls outside the current range, a new class is introduced. Both of these behaviors reflect the concepts of response shift and recalibration, where humans adjust their internal standards over time (Visser et al., 2000; 2005).

## 5 Theoretical Results

In this section, we present some theoretical results to characterize the solution space of the rMSE objective. First, we list our set of assumptions under which our theoretical results hold.

**Assumption 1** (Deterministic Reward Realizability)**.** *The true (unobservable) returns* $\{G^*(\tau_i)\}_{i=1}^K$ *of trajectories* $\{\tau_i\}_{i=1}^K$ *in the dataset* $\mathcal{D}$ *are generated by an underlying (unobservable) deterministic reward function* $r^*$*.*

**Assumption 2** (Binning)**.** *The trajectories are grouped into bins by partitioning the return range* $[\min_i G^*(\tau_i), \max_i G^*(\tau_i)]$ *into* $n$ *rating classes* $\{c_i\}_{i=0}^{n-1}$ *and assigning each trajectory to the class that corresponds to its return. No further assumption about the bin structure is required.*

**Assumption 3** (Hypothesis Class Realizability)**.** $r^*$ *belongs to the hypothesis class.*[3]

**Assumption 4** (Exactness of Differentiable Ranking)**.** *The differentiable ranking function used by rMSE,* $\hat{R}_\theta$*, produces the true rank of each element in a list. For example, given the array* $[1, 5, 2]$*, the function outputs* $[0.0, 2.0, 1.0]$*.*

Note that Assumptions 1 and 2 are dictated by the problem formulation, whereas Assumption 3 is a standard requirement in optimization problems. While Assumption 4 can hold in practice (e.g., using the differentiable sorting method proposed by Blondel et al. (2020) with suitable regularization, as illustrated in Appendix A.4); we relax this assumption later in this section.

---

[2]We confirm this performance degradation in Appendix B, highlighting this relative advantage of rMSE.

[3]In our experiments, we use specific neural network architectures as the hypothesis class, but the theory results are not limited to any specific class of functions.

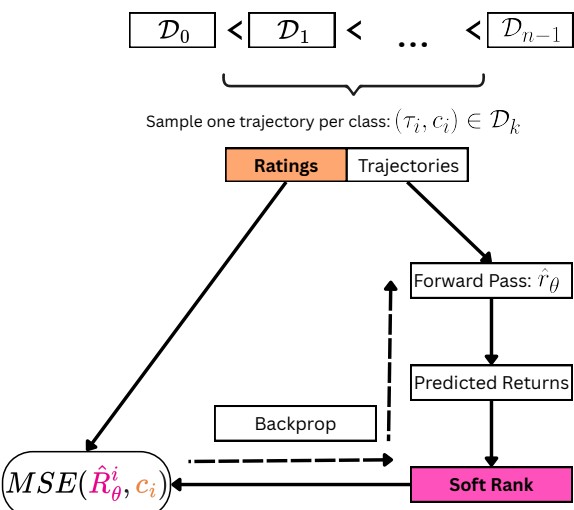

Figure 1: Illustration of the R4 learning process: Given a dataset of trajectory–rating pairs, we compute the predicted return for each trajectory under $\hat{r}_\theta$ and apply a differentiable sorting algorithm to obtain soft ranks. Then, we minimize the MSE between the soft ranks and the original ratings.

Furthermore, the requirement that the ranking operator be differentiable arises solely from the use of a gradient-based optimizer (Kingma & Ba, 2015); it is not required by the optimization objective itself. A non-gradient-based optimizer could be used with hard rankings.

**Definition 1.** *The set of reward functions $\mathcal{R}$ is the set of feasible solutions that satisfy Assumptions 1–3. More formally,*

$$\mathcal{R} \triangleq \{r_\theta | c(\tau_i) < c(\tau_j) \implies G_\theta(\tau_i) < G_\theta(\tau_j) \,\forall \tau_i, \tau_j \in \mathcal{D}\} \tag{2}$$

Note that $\mathcal{R}$ is the set of reward functions in the hypothesis class that we care to find, as they satisfy all assumptions imposed by the problem.

**Proposition 1** (Consistency)**.** *Under Assumptions 1-4, the data-generating reward function $r^*$ is always contained in the solution set of the rMSE objective, but is not guaranteed to be in the solution class of RbRL objective[4]. Formally,*

$$r^* \in \arg\min_\theta \mathcal{L}_{rMSE}(\theta) \quad \text{but} \quad r^* \notin \arg\min_\theta \mathcal{L}_{RbRL}(\theta) \text{ in general.} \tag{3}$$

The proof is given in Appendix A.2.1. Next, in Theorem 1 we show that the solution set of the rMSE objective is complete and minimal under Assumptions 1-4; the set of reward functions, $\mathcal{R}$, induced by Assumptions 1-3 (in Definition 1) is equivalent to the rMSE solution set. In other words, there exists no other objective function that can further reduce the rMSE solution set without introducing additional assumptions. Doing so would risk missing out on some of the potential reward functions in $\mathcal{R}$. This also means that any reward function outside the rMSE solution set is not the data generating reward function, $r^*$.

**Theorem 1** (Completeness and Minimality)**.** *Under Assumptions 1-4, the solution set of rMSE is complete and minimal. Formally,*

$$r_\theta \in \mathcal{R} \iff r_\theta \in \arg\min_\theta \mathcal{L}_{rMSE}(\theta), \quad \forall r_\theta \tag{4}$$

The proof is given in Appendix A.2.2. Theorem 1 establishes the completeness and minimality of rMSE under Assumptions 1–4. However, to ensure that the theorem's guarantees extend to settings where Assumption 4 may fail, we introduce a relaxed version, Assumption 5, and prove an analogous result under the weaker condition in Theorem 2.

---

[4]In the proof, we characterize exactly when $r^*$ will be in the solution class of RbRL objective.

**Assumption 5** (Bounded Ranking Error - Relaxed Assumption 4). *For any element in the input array, the rank predicted by the differentiable sorting operator differs from its true rank by at most $\epsilon$. Formally, for any input vector $\mathbf{v}$ and true ranking function $R$,*

$$\left| \hat{R}(v_i) - R(v_i) \right| \leq \epsilon, \quad \forall i,$$

*where $0 \leq \epsilon < \frac{\sqrt{2n}-2}{n-2}$ is a constant and $n > 2$ is the number of elements in $\mathbf{v}$.*

**Definition 2.** *We define the relaxed solution set for $\mathcal{L}_{rMSE}$ as:*

$$\mathcal{R}_{rMSE} \stackrel{\triangle}{=} \{r_\theta | \mathcal{L}_{rMSE}(\theta) \leq \epsilon^2\} \quad \text{or simply} \quad \mathcal{L}_{rMSE}(\theta) \leq \epsilon^2$$

**Theorem 2** (Completeness and Minimality under Bounded Ranking Error - Relaxed Theorem 1). *Under Assumptions 1-3 and 5, the solution set of rMSE, $\mathcal{R}_{rMSE}$, is complete and minimal. Formally,*

$$r_\theta \in \mathcal{R} \iff r_\theta \in \mathcal{R}_{rMSE}, \quad \forall r_\theta \tag{5}$$

The proof is given in Appendix A.3. Theorem 2 shows that, as long as the error of the ranking operator satisfies the bound in Assumption 5, our guarantees with rMSE still hold.

# 6 EXPERIMENTAL SETUP AND RESULTS

In this section, we first outline the experimental design and results for the offline feedback setting, followed by those for the online feedback setting.

## 6.1 OFFLINE FEEDBACK EXPERIMENTS

**Experimental Setup** We evaluate RbRL and R4 in the offline feedback setting in OpenAI Gym-domains `Reacher`, `Inverted Double Pendulum`, and `Half Cheetah`. In this setting, a reward model is trained on a static dataset of feedback, as opposed to the online feedback setting, where feedback is collected iteratively during RL training. The offline setup avoids additional choices such as the feedback frequency or trajectory sampling method, reducing the number of hyperparameters and allowing us to better isolate the impact of the learning objective. While it also serves to show that R4 extends to offline settings, our primary goal is to demonstrate the performance gains that come specifically from replacing the RbRL loss with the R4 loss, rMSE.

To obtain offline trajectories, we train a Soft Actor-Critic (SAC) (Haarnoja et al., 2018) agent from Stable-Baselines3 (Raffin et al., 2021) using the environment's reward function and store the resulting trajectories along with their ground-truth returns. To systematically evaluate performance, we then use a simulated teacher that assigns scalar ratings to each trajectory based on its ground-truth return. Specifically, we define a set of return thresholds, where each trajectory is labeled according to the return bin it falls into, such that any trajectory $\tau_i$ with return $b[k] \leq G(\tau_i) < b[k+1]$ is assigned rating class $c(\tau_i) = k$. We then construct a balanced dataset by sampling an equal number of trajectories from each class, with $\mathcal{D}_k$ denoting the subset for class $k$ and $\mathcal{D} = \bigcup_{k=0}^{n-1} \mathcal{D}_k$.

Reward models are trained via supervised learning on $\mathcal{D}$, and we evaluate R4 against RbRL under the same training procedure. We also include the environment reward function as a baseline. Full details for reproducibility are provided in Appendix C.4. After training the reward model $\hat{r}_\theta$, we train an RL agent on an unseen environment seed using $\hat{r}_\theta$ as the reward. This process is repeated for 5 random seeds to account for variability in both reward learning and policy optimization. Performance is measured using the environment's ground-truth reward. We report learning curves with individual runs shown in light colors and the mean in a darker color. To test for significant differences between R4 and RbRL, we perform *t*-tests with a significance level of $\alpha = 0.05$.

**Offline Feedback Results** Figure 2 shows that, under otherwise identical conditions, reward functions trained with R4 consistently lead to better downstream RL performance than those learned with RbRL. In particular, when used to train SAC, R4 reward functions led to statistically faster learning in `Inverted Double Pendulum` and `Half Cheetah`, and higher final returns in `Reacher` ($p < 0.05$) as compared to RbRL (see Appendix B.4).

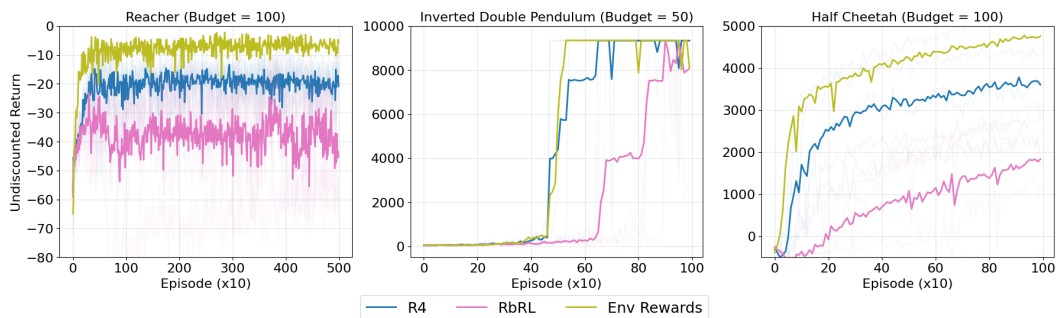

Figure 2: Performance of a SAC agent trained with (1) R4, (2) RbRL, and (3) the environment reward. `Budget` denotes the number of labeled trajectories used to learn the offline reward function.

## 6.2 ONLINE FEEDBACK EXPERIMENTS

**Experimental Setup** In our online experiments, a SAC agent interacts with the environment and learns from the estimated reward function $\hat{r}_\theta$. From these interactions, trajectory segments are periodically sampled and rated by a simulated teacher, which provides feedback according to the environment's ground-truth reward. This feedback is then used to update the reward model, guiding the agent's future behavior. We evaluate R4 against RbRL and 3 preference learning algorithms: PEBBLE (Lee et al., 2021), SURF (Park et al., 2022), and QPA (Hu et al., 2024), across 6 DMC environments: `Walker-walk`, `Walker-stand`, `Cheetah-run`, `Quadruped-walk`, `Quadruped-run`, and `Humanoid-stand`. For all methods, we fix a feedback budget: in rating-based approaches, it counts rated trajectories, while in preference-based approaches, it counts pairwise comparisons. Since each comparison involves two trajectories, the same budget requires the teacher to assess twice as many trajectories in preference-based methods.

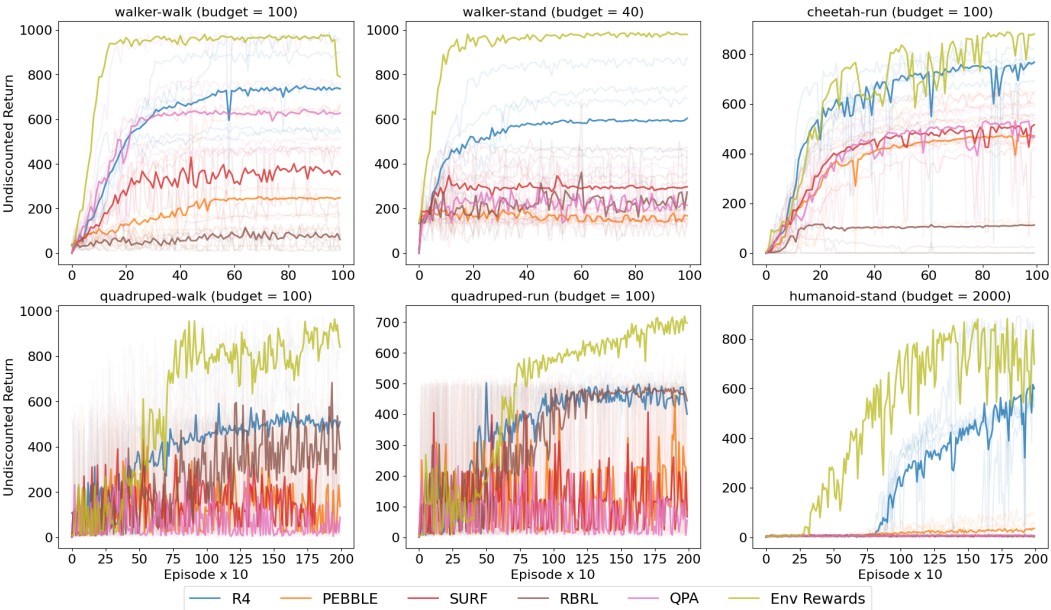

Figure 3: This shows online SAC performance evaluated with the true environment reward, using either the environment reward or learned rewards from different rating/preference-based algorithms.

All implementation details are provided in Appendix C.3. For all online experiments, we follow the standard SAC implementation from PEBBLE (Lee et al., 2021). Regarding the reward learning components, the baselines use their default configurations: a uniform feedback schedule and their respective trajectory sampling strategies (uncertainty-based for all except QPA, which uses a near

on-policy strategy). For completeness, we tested the baselines with our dynamic feedback schedule; however, this degraded their performance (see Appendix B, Figure 11). Therefore, we report results using each method's strongest configuration. As in the prior section, performance is measured using the environment's ground-truth reward, with learning curves showing 5 individual runs (light) and their mean (dark). To test for significant differences between R4 and the baselines, we perform $t$-tests ($\alpha = 0.05$) with Bonferroni correction. As we conduct 4 comparisons per environment, we use a corrected significance threshold of $\frac{\alpha}{4} = 0.0125$ to control the family-wise error rate.

**Online Feedback Results** Figure 3 shows that R4 consistently matches or outperforms existing approaches across all tested environments. In particular, using R4, the SAC agent learns significantly faster than all baselines in 3 of the 6 environments and achieves higher final returns in 4 of the 6 environments ($p < 0.0125$). See Appendix B.4 for detailed information on the $t$-test results.

## 6.3 ABLATIONS

To assess the impact of our design choices, specifically the dynamic feedback schedule and the stratified trajectory sampling strategy, we ablate these components and evaluate performance on `Walker-walk`, `Cheetah-run`, and `Quadruped-walk` in the online feedback setting. As shown in Figure 4, R4 performs comparably without these additions (red curves) in 2 out of 3 tested domains. Furthermore, using either technique on its own is sufficient to sustain performance, while combining both yields small but consistent improvements. Moreover, Appendix B (Figures 12 and 13) demonstrates R4's robustness: it maintains strong performance under high levels of noisy feedback (80% noise) compared to RbRL at only 10% noise, and it achieves consistent results across different numbers of rating classes, whereas RbRL is sensitive to the choice of class count.

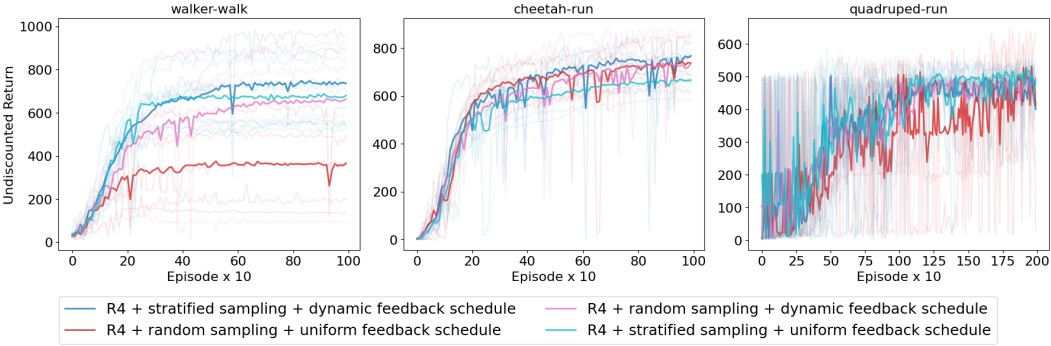

Figure 4: We evaluate R4 under ablations of our implementation choices: stratified sampling and the dynamic feedback schedule. We find that both features can improve the base R4 method.

## 7 CONCLUSION

Reward design remains a fundamental challenge in RL. While preference-based RL has been the dominant approach for learning rewards from human feedback, rating-based approaches have recently emerged as a promising alternative. By allowing humans to evaluate behaviors individually, ratings may reduce cognitive load and enable richer supervision than pairwise preferences (White et al., 2024). Therefore, in this work, we propose R4, a theoretically grounded algorithm for learning reward functions from multi-class human ratings. Unlike prior work, R4 treats ratings as ordinal feedback and optimizes a rank-based mean squared error loss, allowing the reward model to better exploit the rating structure in the labeled trajectories. To demonstrate the utility of R4, we first provide a theoretical analysis showing that it yields minimal and complete solutions under mild assumptions. Next, we empirically demonstrate its effectiveness across both offline and online feedback scenarios. In particular, R4 can outperform both rating- and preference-based RL baselines on robotic locomotion tasks, producing reward models that lead to more performant policies. Overall, our results represent an important step toward reward learning methods that maintain theoretical rigor while efficiently leveraging teacher feedback.

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

APPENDIX

# A THEORY RESULTS

## A.1 DERIVATIVE OF RbRL LOSS FUNCTION

The RbRL White et al. (2024) loss function is defined as:

$$\mathcal{L}_{\text{RbRL}} = \mathbb{E}_\tau \left( - \sum_{i=0}^{N-1} \mu_i \log(Q(\tau_i)) \right) \tag{6}$$

Where $\mu_i$ is the indicator function for the assigned label, ie. $\mu_i = 1$ when the trajectory $\tau_i$ is assigned the label class $i$ in the dataset, and $0$ otherwise. Furthermore, the function $Q$ is defined as:

$$Q(\tau_i) = \frac{e^{-k(\hat{G}_i - B_i)(\hat{G}_i - B_{i+1})}}{\sum_j e^{-k(\hat{G}_j - B_j)(\hat{G}_j - B_{j+1})}} \tag{7}$$

Here, $\{B_i\}_{i=0}^{N-1}$ are class decision boundaries and $k$ is a hyperparameter. We write $\hat{G}_i$ instead of $\hat{G}_\theta(\tau_i)$ for convenience to denote the the normalized predicted return. The derivative of this loss is:

$$\frac{\partial \mathcal{L}_{\text{RbRL}}}{\partial \hat{G}_i} = \mathbb{E}_\tau \left( k \sum_j (\mu_j - Q(\tau_i))(2\hat{G}_i - B_j - B_{j+1}) \right) \tag{8}$$

This result is not new and is presented only because it is used in the proof of proposition 1.

## A.2 PROOFS

### A.2.1 PROOF FOR PROPOSITION 1

**Part 1:** Since $r^*$ is the deterministic data generating reward function, $c(\tau_i) < c(\tau_j) \implies G^*(\tau_i) < G^*(\tau_j)$, where $G^*$ is the trajectory return using the reward function $r^*$. Furthermore, If we try to rank the trajectories according to their corresponding $G^*$, we will recover their $c(\tau_i)$. Hence, $\hat{R}_\theta(\tau_i) = c(\tau_i)$ for all $i$ when $\hat{r}_\theta = r^*$. This implies that $\mathcal{L}_{\text{rMSE}}(\theta) = 0$ when $\hat{r}_\theta = r^*$. Hence $r^* \in \arg\min_\theta \mathcal{L}_{\text{rMSE}}(\theta)$.

**Part 2:** To show that $r^* \notin \arg\min_\theta \mathcal{L}_{\text{RbRL}}(\theta)$ in general, providing a counterexample suffices. Equation 8 shows that $\mathcal{L}_{\text{RbRL}}$ is minimized when the return for each trajectory in a rating class is exactly equal to either $B_i$, $B_{i+1}$, or $\frac{B_i + B_{i+1}}{2}$ White et al. (2024). Consider the reward function $r^*$ to assign the return of $\frac{B_i + B_{i+1}}{2} + \epsilon$ for some $0 < \epsilon < \frac{B_{i+1} - B_i}{2}$ to each trajectory in the same rating class. Such $r^*$ would not belong to the solution class of $\mathcal{L}_{\text{RbRL}}$. Hence, $r^* \notin \arg\min_\theta \mathcal{L}_{\text{RbRL}}(\theta)$ in general. $\qquad\square$

### A.2.2 PROOF OF THEOREM 1

First, let us assume that there exists some reward function $r_\theta \in \mathcal{R}$.

Since $r_\theta \in \mathcal{R}$,

$$c(\tau_i) < c(\tau_j) \implies G_\theta(\tau_i) < G_\theta(\tau_j)$$

Therefore, if we pick one trajectory from each rating class (without loss of generality):

$$c(\tau_0) < c(\tau_1) < \cdots < c(\tau_{n-1}), \quad \text{where } c(\tau) \in \{0, 1, \cdots, n-1\}$$
$$\implies G_\theta(\tau_0) < G_\theta(\tau_1) < \cdots < G_\theta(\tau_{n-1})$$

This implies that the predicted ranks $\{R_\theta(\tau_i)\}_{i=0}^{n-1}$ of the $\{G_\theta(\tau_i)\}_{i=0}^{n-1}$ will also follow the same order:

$$R_\theta(\tau_0) < R_\theta(\tau_1) < \cdots < R_\theta(\tau_{n-1}), \quad \text{where } R_\theta \in \{0, 1, \cdots, n-1\} \tag{9}$$

$$\implies \frac{1}{n} \sum_{i=0}^{n-1} (c(\tau_i) - R_\theta(\tau_i))^2 = 0, \quad \text{Both } c \text{ and } R_\theta \text{ are distinct integers in [0, n-1] (4)} \tag{10}$$

$$\implies r_\theta \in \arg\min_\theta \mathcal{L}_{\text{rMSE}}(\theta) \tag{11}$$

Therefore, $r_\theta \in \mathcal{R} \implies r_\theta \in \arg\min_\theta \mathcal{L}_{\text{rMSE}}(\theta)$.

Now, let us assume that there exists a reward function $r_\theta \in \arg\min_\theta \mathcal{L}_{\text{rMSE}}(\theta)$.

Let us pick one trajectory from each bin (without loss of generality):

$$\implies c(\tau_0) < c(\tau_1) < \cdots < c(\tau_{n-1})$$

Now, since $r_\theta \in \arg\min_\theta \mathcal{L}_{\text{rMSE}}(\theta)$,

$$\frac{1}{N} \sum_{i=0}^{n-1} (c(\tau_i) - R_\theta(\tau_i))^2 = 0$$

$$\implies c(\tau_i) = R_\theta(\tau_i), \forall i$$

$$\implies R_\theta(\tau_0) < R_\theta(\tau_1) < \cdots < R_\theta(\tau_{n-1})$$

$$\implies G_\theta(\tau_0) < G_\theta(\tau_1) < \cdots < G_\theta(\tau_{n-1})$$

Therefore, $c(\tau_i) < c(\tau_j) \implies G_\theta(\tau_i) < G_\theta(\tau_j)$, which implies that $r_\theta \in \mathcal{R}$. Hence, using both of these results, $r_\theta \in \mathcal{R} \iff r_\theta \in \arg\min_\theta \mathcal{L}_{\text{rMSE}}(\theta)$ $\qquad\square$

### A.3 RELAXING ASSUMPTION 4

*Proof of Theorem 2.*
The proof follows a similar structure as the proof for Theorem 1.

First, suppose that there exists some reward function $r_\theta \in \mathcal{R}$.

Since $r_\theta \in \mathcal{R}$, if we pick one trajectory from each rating class (without loss of generality):

$$c(\tau_0) < c(\tau_1) < \cdots < c(\tau_{n-1}), \quad \text{where } c(\tau) \in \{0, 1, \cdots, n-1\}$$

$$\implies G_\theta(\tau_0) < G_\theta(\tau_1) < \cdots < G_\theta(\tau_{n-1})$$

Since $\{G_\theta(\tau_i)\}_{i=0}^{n-1}$ follow the same order as $\{c(\tau_i)\}_{i=0}^{n-1}$, and the predicted ranks $\{R_\theta(\tau_i)\}_{i=0}^{n-1}$ differs from the true ranks by at most $\epsilon$,

$$\implies \frac{1}{n} \sum_{i=0}^{n-1} (R_\theta(\tau_i) - c(\tau_i))^2 \leq \epsilon^2$$

$$\implies \mathcal{L}_{\text{rMSE}}(\theta) \leq \epsilon^2$$

$$\implies r_\theta \in \mathcal{R}_{\text{rMSE}}$$

Now, let us assume that there exists a reward function $r_\theta \in \mathcal{R}_{\text{rMSE}}$:

If we pick one trajectory from each rating class (without the loss of generality), then:

$$c(\tau_0) < c(\tau_1) < \cdots < c(\tau_{n-1})$$

Since $r_\theta \in \mathcal{R}_{\text{rMSE}}$,

$$\implies \mathcal{L}_{\text{rMSE}}(\theta) \leq \epsilon^2 \tag{12}$$

$$\implies \frac{1}{n} \sum_{i=0}^{n-1} \left( R_\theta(\tau_i) - c(\tau_i) \right)^2 \leq \epsilon^2 \tag{13}$$

To conclude the proof, we must show that the predicted returns $\{G_\theta(\tau_i)\}_{i=0}^{n-1}$ preserve the same ordering as the class labels $\{c(\tau_i)\}_{i=0}^{n-1}$. If the ordering is preserved, then $r_\theta \in \mathcal{R}$ immediately follows.

Consider the cases where the ordering is violated because of the reward function[5]. For $\epsilon < 0.5$, the "least harmful" violation (i.e., the one that produces the smallest possible $\mathcal{L}_{\text{rMSE}}$ while still breaking the ordering because of the returns from the reward function) occurs when exactly two adjacent elements swap their order, while all other predictions are correct. Without loss of generality, suppose these elements are at indices $k$ and $k + 1$, and that $R_\theta(\tau_i) = c(\tau_i)$ for all $i \notin \{k, k + 1\}$.

For example, if the true class labels are $[0, 1, 2]$, then the smallest-error misordering happens when the predictions are $[1 - \epsilon, 0 + \epsilon, 2]$: the first two items are swapped but deviate from their true labels by only $\epsilon$.

This case gives the minimum possible $\mathcal{L}_{\text{rMSE}}$ under an incorrect ordering, which equals $\frac{2(1-\epsilon)^2}{n}$, determined as follows:

$$\frac{1}{n} \sum_{i=0}^{n-1} \left( R_\theta(\tau_i) - c(\tau_i) \right)^2 = \frac{1}{n} \left[ \underbrace{\sum_{i=0}^{k-1} \left( R_\theta(\tau_i) - c(\tau_i) \right)^2}_{=0} + (R_\theta(\tau_k) - c(\tau_k))^2 \right.$$

$$\left. + (R_\theta(\tau_{k+1}) - c(\tau_{k+1}))^2 + \underbrace{\sum_{i=k+2}^{n-1} \left( R_\theta(\tau_i) - c(\tau_i) \right)^2}_{=0} \right]$$

$$= \frac{(R_\theta(\tau_k) - c(\tau_k))^2 + (R_\theta(\tau_{k+1}) - c(\tau_{k+1}))^2}{n}$$

$$= \frac{(1 - \epsilon - 0)^2 + (\epsilon - 1)^2}{n}$$

$$= \frac{2(1 - \epsilon)^2}{n}. \tag{14}$$

Therefore, to exclude such incorrect orderings from the relaxed solution set $\mathcal{R}_{\text{rMSE}}$, we require that the lower bound on the error of an invalid solution exceed $\epsilon^2$. More specifically:

$$\frac{2(1 - \epsilon)^2}{n} > \epsilon^2$$

$$\implies 0 \leq \epsilon < \frac{\sqrt{2n} - 2}{n - 2}$$

which is the bound on $\epsilon$ in assumption 5. Now, since $\epsilon$ satisfies this bound, continuing from 13, we can be sure that:

$$G_\theta(\tau_0) < G_\theta(\tau_1) < \cdots < G_\theta(\tau_{n-1})$$

$$\implies r_\theta \in \mathcal{R}$$

---

[5]Since we only want to exclude the reward functions where the returns from the reward function do not follow the ordering, we consider the cases with $\epsilon < 0.5$. Otherwise the ranking function loses its meaning and we start misordering things because of the ranking function

Combining these two results, we have shown that under assumptions 1, 2, 3 and 5:

$$r_\theta \in \mathcal{R} \iff r_\theta \in \mathcal{R}_{\text{rMSE}}$$

$\square$

### A.4 SAMPLE OUTPUTS FROM FAST-SOFT RANK

Here, we present a few outputs from the fast-soft ranking algorithm, to justify Assumption 4. With an appropriate value of `regularization_strength`, the ranking operator outputs the true ranks of all elements in most cases.

```python
for _ in range(10):
    x = np.random.uniform(0,10,10)
    print(soft_rank(x, regularization_strength=0.01))

'''
Outputs:
[ 4.   5. 10.   3.   6.   2.   9.   1.   7.   8.]
[ 9.   4. 10.   8.   2.   1.   7.   3.   5.   6.]
[ 5.   7.   6.   9.   4.   2.   3.   8. 10.   1.]
[ 8.   4. 10.   6.   7.   5.   1.   3.   9.   2.]
[ 8.   5.   6.   4.   1.   7.   9.   3. 10.   2.]
[ 7.   5.   9.   1.   2.   6. 10.   4.   8.   3.]
[ 5.   8.   9.   3.   2. 10.   1.   7.   6.   4.]
[ 1.   7. 10.   4.   9.   3.   6.   5.   8.   2.]
[ 7.   9. 10.   5.   6.   1.   3.   8.   4.   2.]
[ 8.   9.   4.   7. 10.   3.   6.   2.   5.   1.]
'''
```

Listing 1: Sample Outputs From Soft Rank

## B ADDITIONAL RESULTS

### B.1 HUMAN STUDIES

In this section, we describe our human-subject pilot study. We conducted the study with five participants (two authors/experts and three non-authors/non-experts). Each rater was shown trajectories from OpenAI Gym's `reacher` environment sequentially and asked either to rate each trajectory or skip it. Each rater could decide how many bins to use, choosing any value between 3 and 10. The raters were asked to collect ratings for 100-200 trajectories in total. After collecting the data, we trained an offline reward function using 100 randomly selected trajectories from each participant's dataset. Using the learned reward function, we then trained a SAC agent on the same environment. We repeated this process five times to account for randomness in reward-function learning and SAC training.

We report the aggregated learning curves in Figure 5. The mean performance of the policies derived from each individual rater is shown in lighter colors, while the average across all raters is shown in darker colors. These results indicate that R4 performs better than RbRL despite any rater-specific biases and inconsistencies that may have occurred. Furthermore, on average, R4 with human ratings performs similarly to R4 with perfect simulated ratings.

Finally, Figures 6 through 10 show the SAC learning curves for each rater's reward function (left). Different seeds are shown in lighter colors, while the mean is shown in a darker color. The middle plot shows each rater's distribution of labels. It reveals that (i) the distributions are highly non-uniform for all raters, and (ii) the distributions vary substantially across raters, with different individuals using different numbers of rating classes. The right plot shows the distribution of undiscounted environment returns associated with each label in the rated dataset. This confirms that the human ratings are highly imperfect, with substantial overlap in true returns across labels. These results further support our claim that R4 is robust to rater bias and variance.

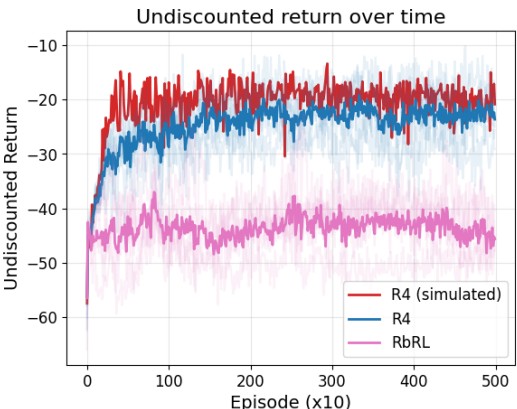

Figure 5: Combined performance of policies trained on reward models learned from five human raters. Light curves show the average performance across five SAC training runs for each rater; the dark curve shows the overall mean across raters. R4 consistently outperforms RbRL despite substantial variation in individual rating behavior. Furthermore, on average, R4 with human ratings performs similarly to R4 with perfect simulated ratings.

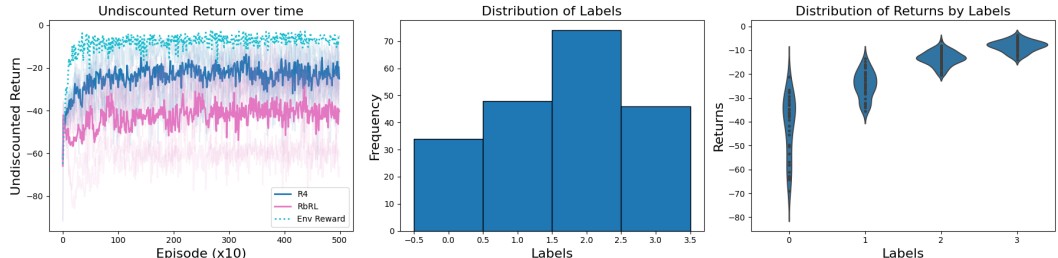

Figure 6: (Left) Learning curves for Participant 1's reward function, with individual random seeds shown in lighter colors and the mean in darker color. (Middle) Histogram of Participant 1's labeling distribution. (Right) Violin plot of undiscounted environment returns conditioned on each label, showing substantial overlap in returns and highlighting the noisiness and imperfection of the ratings.

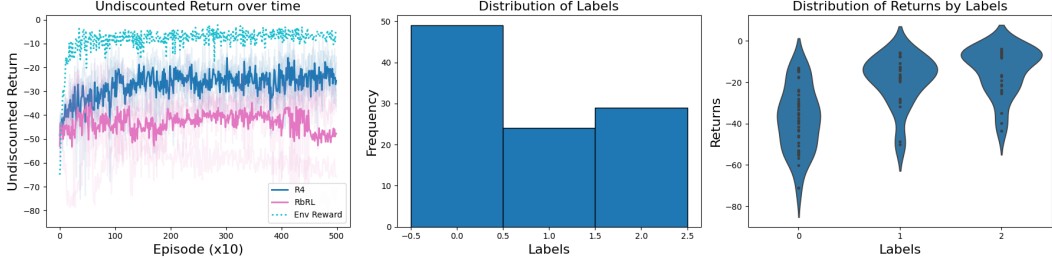

Figure 7: Participant 2's results and data distribution.

## B.2 SIMULATED EXPERIMENTS

To assess the impact of the dynamic feedback schedule and sampling tricks on the baselines, we tested them with these modifications included. Figure 11 shows that the baselines' performance either remains similar or degrades compared to Figure 3.

Second, we evaluate the resilience of R4 to noisy feedback on the `Inverted Double Pendulum` task in the offline setting, where both R4 and RbRL achieve similar final performance under noiseless conditions. We focus on the offline setting because it isolates the effect of noise on

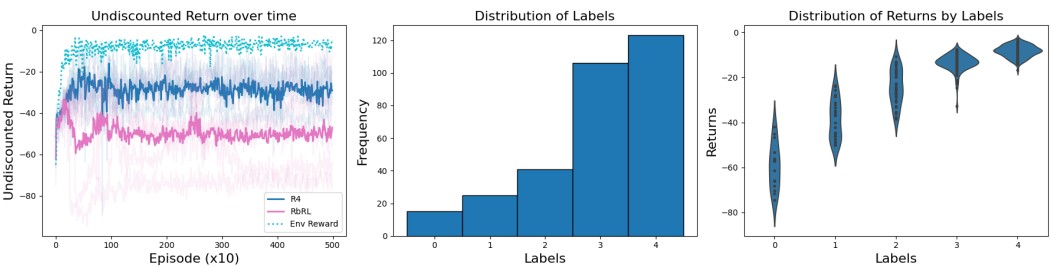

Figure 8: Participant 3's results and data distribution.

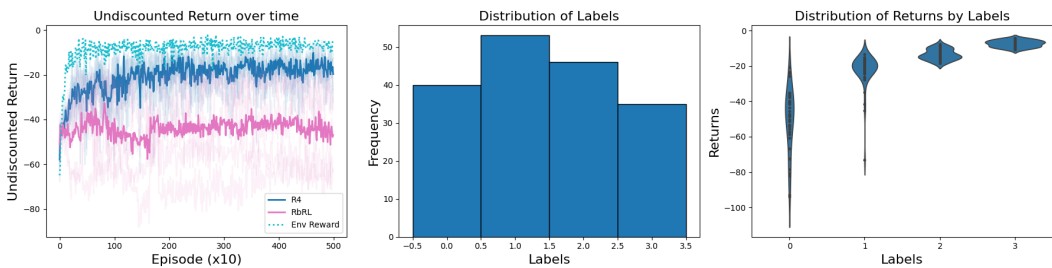

Figure 9: Participant 4's results and data distribution.

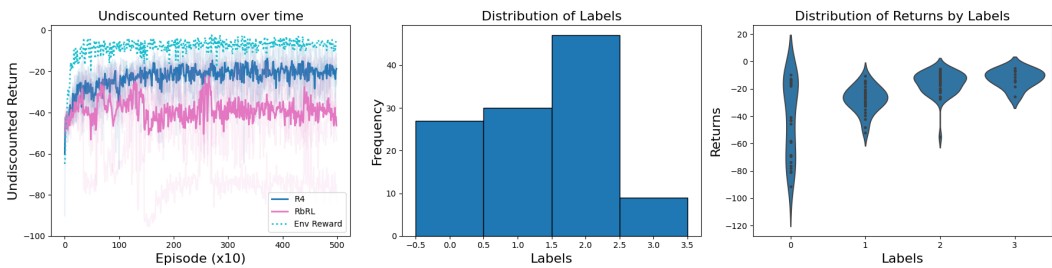

Figure 10: Participant 5's results and data distribution.

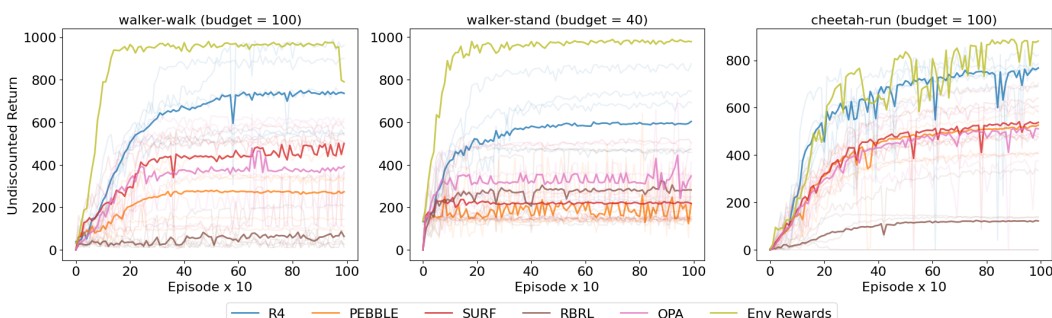

Figure 11: Mean undiscounted return (computed using the environment's reward function) versus the number of episodes when training a SAC agent with R4 and various baselines in the online setting. The baselines are allowed to use our dynamic feedback schedule and their respective query sampling tricks.

reward learning. To simulate noisy human feedback, we randomly select $\eta\%$ of the trajectories in the dataset $\mathcal{D}$ and reassign them to $\texttt{true\_bin} \pm 1$ with probability $0.5$ each.

Figure 12a shows R4 performance under varying noise levels. While performance naturally decreases as $\eta$ increases, R4 remains robust even at high noise levels. Figure 12b compares R4 with

RbRL under the same conditions, showing that RbRL fails even at small noise levels. Notably, R4 with 80% noise achieves performance comparable to RbRL with only 10% noise.

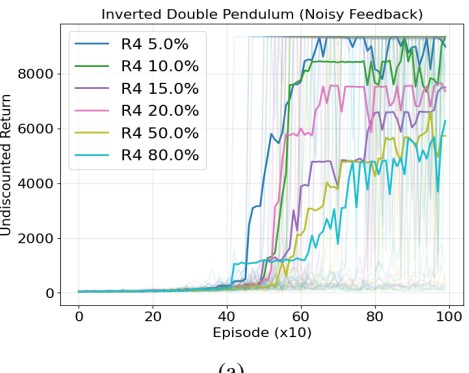
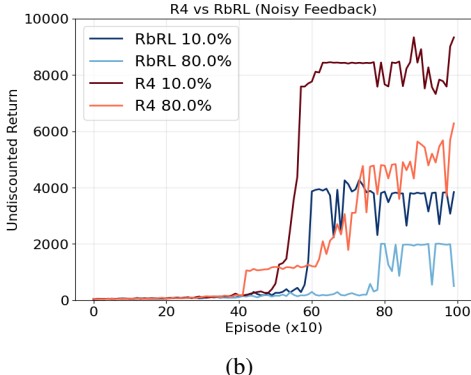

(a)                                                    (b)

Figure 12: (a) R4 objective under varying levels of noise. (b) Comparison of R4 (reds) and RbRL (blues) under different noise levels.

Furthermore, we study the impact of the number of rating classes on R4 in the `Reacher` environment. Figure 13 shows that although RbRL's performance depends significantly on the number of bins, R4 remains consistent.

Finally, we study the impact of the fast-soft-ranking (Blondel et al., 2020) regularization strength in R4 for the `Inverted Double Pendulum` environment. In Figure 15, we plot the undiscounted return averaged over the last 100 episodes as a function of the regularization strength. The plot shows that 83% of the runs with regularization strengths between 0.065 and 1 learn a successful policy.

Overall, these results highlight that R4 is robust to dynamic feedback schedules, resilient to noisy feedback, and largely insensitive to the choice of rating classes, in contrast to RbRL, which is sensitive to all three.

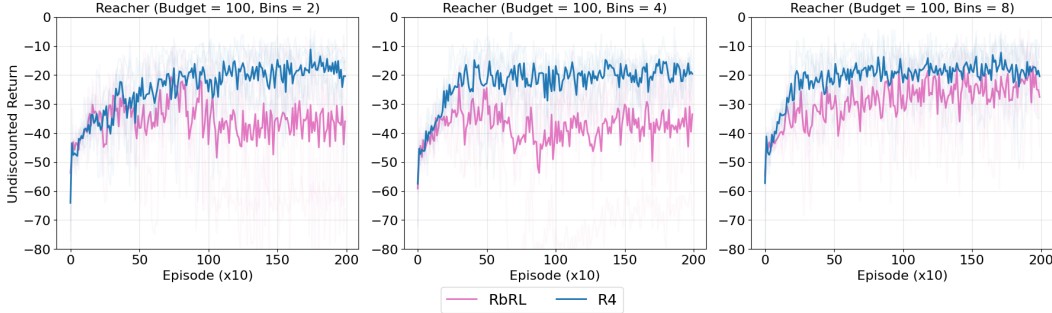

Figure 13: Undiscounted return vs number of episodes for reacher with varying number of bins.

### B.3 QUALITY OF LEARNED REWARD FUNCTIONS

To assess the quality of the reward functions learned by R4 relative to the baselines, we first present a scatter plot comparing undiscounted returns from the learned reward functions against the undiscounted environment returns encountered during a single online run (Figure 14). We show this for three environments and for all methods. The plots indicate that R4 consistently captures a meaningful relationship between learned and actual returns across environments.

Furthermore, to evaluate reward quality quantitatively, we report the Trajectory Alignment Coefficient (TAC) Muslimani et al. (2025) in Table 1. TAC is a reward alignment metric that measures how similarly two reward functions rank a set of trajectories, where a TAC of 1 indicates perfect alignment and a TAC of 1 indicates perfect negative correlation. We compare the reward functions

learned by each method with the ground truth reward functions using TAC. For this comparison, we consider the trajectories encountered during training. Table 1 shows that R4 produces reward functions that are more aligned to the ground-truth reward in two out of three environments.

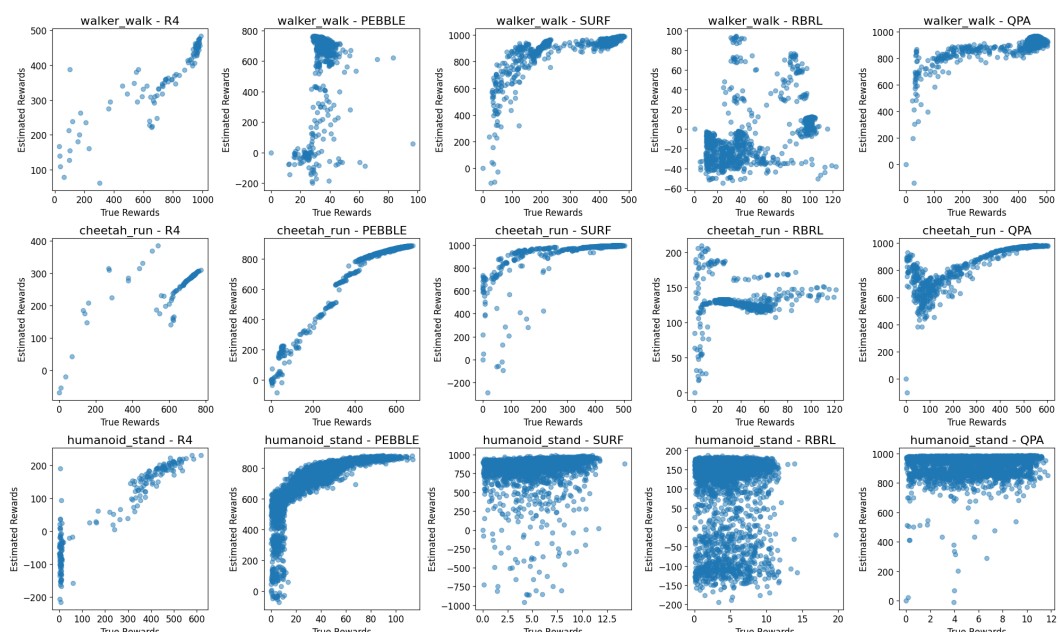

Figure 14: Qualitative results

| Environment | PEBBLE | SURF | RbRL | QPA | R4 |
|---|---|---|---|---|---|
| walker_walk | 0.4681 | 0.5386 | 0.2643 | 0.6521 | 0.7168 |
| cheetah_run | 0.8828 | 0.8366 | 0.0200 | 0.8107 | 0.5956 |
| humanoid_stand | 0.3537 | 0.1640 | -0.0116 | 0.1452 | 0.6312 |

Table 1: Average TAC scores across environments and methods.

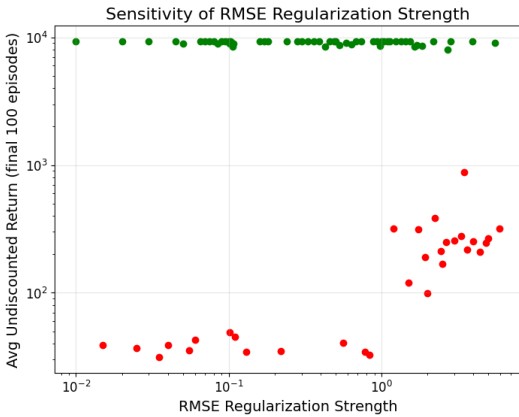

Figure 15: Regularization strength of fast-soft ranking (Blondel et al., 2020) vs final learned policy return for Inverted Double Pendulum.

## B.4 STATISTICAL SIGNIFICANCE

To assess the statistical significance of our main results presented in Figures 2 and 3, we applied Welch's t-test to two key metrics: the average return over the last 100 episodes and the area under

the learning curve (AUC). These metrics capture both the ultimate performance and the overall learning dynamics of each method across multiple random seeds. Table 2 reports the results for the offline feedback runs shown in Figure 2, while Tables 3–6 present analogous results for the online feedback setting in Figure 3. Across nearly all environments, our method demonstrates statistically significant improvements over the baselines in both final return and AUC, indicating not only higher ultimate performance but also more efficient learning.

| Environment | Metric | R4 (Mean $\pm$ SD) | RbRL (Mean $\pm$ SD) | t-stat | p-value |
|---|---|---|---|---|---|
| Reacher | Return | $-19.91 \pm 3.45$ | $-39.13 \pm 13.03$ | 2.85 | 0.0398 |
| | AUC | $-10331.49 \pm 1448.88$ | $-18863.86 \pm 6327.05$ | 2.63 | 0.0527 |
| Inverted DP | Return | $9212.41 \pm 272.56$ | $8485.79 \pm 1239.53$ | 1.15 | 0.3108 |
| | AUC | $441942.60 \pm 58656.92$ | $206676.88 \pm 82616.80$ | 4.64 | 0.0022 |
| Half Cheetah | Return | $3638.23 \pm 1115.80$ | $1746.10 \pm 1380.16$ | 2.13 | 0.0671 |
| | AUC | $282021.30 \pm 98623.15$ | $75385.09 \pm 90477.05$ | 3.09 | 0.0151 |

Table 2: Comparison of final return and AUC between R4 and Baseline with Welch's t-test results.

| Environment | Metric | R4 (Mean $\pm$ SD) | PEBBLE (Mean $\pm$ SD) | p-value |
|---|---|---|---|---|
| walker-walk | Return | $736.78 \pm 173.37$ | $246.13 \pm 213.81$ | 0.000 |
| | AUC | $59428.38 \pm 11858.29$ | $18832.11 \pm 15324.24$ | 0.003 |
| walker-stand | Return | $594.24 \pm 223.85$ | $154.46 \pm 31.18$ | 0.000 |
| | AUC | $52901.37 \pm 17831.47$ | $16072.85 \pm 1370.81$ | 0.014 |
| cheetah-run | Return | $747.68 \pm 61.25$ | $463.69 \pm 187.02$ | 0.000 |
| | AUC | $59464.45 \pm 3505.70$ | $36415.47 \pm 13841.43$ | 0.027 |
| quadruped-walk | Return | $500.64 \pm 197.69$ | $107.07 \pm 184.75$ | 0.000 |
| | AUC | $76601.70 \pm 28181.97$ | $25782.51 \pm 7008.07$ | 0.021 |
| quadruped-run | Return | $451.30 \pm 85.90$ | $214.81 \pm 234.10$ | 0.000 |
| | AUC | $70808.57 \pm 14600.77$ | $26447.09 \pm 6189.97$ | 0.002 |
| humanoid-stand | Return | $537.04 \pm 157.46$ | $30.30 \pm 32.46$ | 0.000 |
| | AUC | $43349.13 \pm 16132.70$ | $3171.91 \pm 2630.79$ | 0.007 |

Table 3: Comparison of final return and AUC between R4 and PEBBLE across environments with Welch's t-test p-values.

| Environment | Metric | R4 (Mean $\pm$ SD) | SURF (Mean $\pm$ SD) | p-value |
|---|---|---|---|---|
| walker-walk | Return | $736.78 \pm 173.37$ | $366.89 \pm 205.47$ | 0.000 |
| | AUC | $59428.38 \pm 11858.29$ | $30528.74 \pm 16237.83$ | 0.023 |
| walker-stand | Return | $594.24 \pm 223.85$ | $292.50 \pm 114.15$ | 0.000 |
| | AUC | $52901.37 \pm 17831.47$ | $28487.92 \pm 10485.57$ | 0.053 |
| cheetah-run | Return | $747.68 \pm 61.25$ | $496.27 \pm 105.96$ | 0.000 |
| | AUC | $59464.45 \pm 3505.70$ | $39690.18 \pm 8243.98$ | 0.006 |
| quadruped-walk | Return | $500.64 \pm 197.69$ | $142.08 \pm 193.82$ | 0.000 |
| | AUC | $76601.70 \pm 28181.97$ | $28346.64 \pm 5380.05$ | 0.025 |
| quadruped-run | Return | $451.30 \pm 85.90$ | $119.98 \pm 177.57$ | 0.000 |
| | AUC | $70808.57 \pm 14600.77$ | $23723.68 \pm 3381.69$ | 0.002 |
| humanoid-stand | Return | $537.04 \pm 157.46$ | $4.51 \pm 2.73$ | 0.000 |
| | AUC | $43349.13 \pm 16132.70$ | $1113.39 \pm 67.12$ | 0.006 |

Table 4: Comparison of final return and AUC between R4 and SURF across environments with Welch's t-test p-values.

## C  IMPLEMENTATION DETAILS

This section includes the details necessary to replicate our results. Code will be released if the paper is accepted.

| Environment | Metric | R4 (Mean $\pm$ SD) | RBRL (Mean $\pm$ SD) | p-value |
|---|---|---|---|---|
| walker-walk | Return | $736.78 \pm 173.37$ | $77.18 \pm 60.08$ | 0.000 |
| | AUC | $59428.38 \pm 11858.29$ | $6359.18 \pm 3429.20$ | 0.000 |
| walker-stand | Return | $594.24 \pm 223.85$ | $238.70 \pm 132.50$ | 0.000 |
| | AUC | $52901.37 \pm 17831.47$ | $21555.53 \pm 8995.78$ | 0.020 |
| cheetah-run | Return | $747.68 \pm 61.25$ | $111.43 \pm 210.31$ | 0.000 |
| | AUC | $59464.45 \pm 3505.70$ | $9548.40 \pm 16798.53$ | 0.003 |
| quadruped-walk | Return | $500.64 \pm 197.69$ | $437.97 \pm 338.88$ | 0.267 |
| | AUC | $76601.70 \pm 28181.97$ | $51671.65 \pm 24218.54$ | 0.217 |
| quadruped-run | Return | $451.30 \pm 85.90$ | $467.09 \pm 27.19$ | 0.225 |
| | AUC | $70808.57 \pm 14600.77$ | $64727.11 \pm 8174.64$ | 0.493 |
| humanoid-stand | Return | $537.04 \pm 157.46$ | $4.91 \pm 3.09$ | 0.000 |
| | AUC | $43349.13 \pm 16132.70$ | $928.10 \pm 63.57$ | 0.006 |

Table 5: Comparison of final return and AUC between R4 and RBRL across environments with Welch's t-test p-values.

| Environment | Metric | R4 (Mean $\pm$ SD) | QPA (Mean $\pm$ SD) | p-value |
|---|---|---|---|---|
| walker-walk | Return | $736.78 \pm 173.37$ | $627.46 \pm 199.33$ | 0.005 |
| | AUC | $59428.38 \pm 11858.29$ | $54360.80 \pm 17751.89$ | 0.649 |
| walker-stand | Return | $594.24 \pm 223.85$ | $214.50 \pm 101.02$ | 0.000 |
| | AUC | $52901.37 \pm 17831.47$ | $21386.02 \pm 8224.46$ | 0.020 |
| cheetah-run | Return | $747.68 \pm 61.25$ | $501.78 \pm 112.86$ | 0.000 |
| | AUC | $59464.45 \pm 3505.70$ | $39251.21 \pm 6357.56$ | 0.001 |
| quadruped-walk | Return | $500.64 \pm 197.69$ | $58.50 \pm 138.00$ | 0.000 |
| | AUC | $76601.70 \pm 28181.97$ | $12646.85 \pm 5822.67$ | 0.009 |
| quadruped-run | Return | $451.30 \pm 85.90$ | $62.94 \pm 119.64$ | 0.000 |
| | AUC | $70808.57 \pm 14600.77$ | $14861.70 \pm 8587.97$ | 0.000 |
| humanoid-stand | Return | $537.04 \pm 157.46$ | $6.04 \pm 3.04$ | 0.000 |
| | AUC | $43349.13 \pm 16132.70$ | $1115.85 \pm 131.49$ | 0.006 |

Table 6: Comparison of final return and AUC between R4 and QPA across environments with Welch's t-test p-values.

## C.1 Batch Updates

While computing the loss using a single sampled trajectory per class dataset $\mathcal{D}_k$ provides a valid training signal, it can lead to a biased gradient estimate and hinder learning. To improve stability, we perform the soft ranking procedure $B$ times per update step. In each iteration, we sample one trajectory per class dataset, compute predicted returns using $\hat{r}_\theta$, and apply the differentiable sorting algorithm (Blondel et al., 2020) to obtain soft ranks. The resulting $B$ soft rank vectors (of size $n$) are then stacked to form a stacked soft ranks matrix. Correspondingly, we stack the class labels associated with each sampled trajectory into a ratings matrix. We compute the rMSE loss as the mean squared error between the stacked soft ranks and the stacked ratings.

## C.2 Possible Regularization

### C.2.1 L2 Regularization

For training our reward functions, we use an L2 regularization loss (with coefficient $\beta$) defined as:
$$\mathcal{L}_{L2} = \mathbb{E}_{\tau_i}\left[|\hat{r}_\theta(\tau_i)|^2\right]$$

### C.2.2 Out Of Distribution Regularization

Even though we do not use OOD regularization in our experiments, offline RL literature (Kumar et al., 2020; Li et al., 2021) tells that it might be a good tool to have when learning from an under-specified dataset. The idea is to penalize high predicted rewards (under $\hat{r}_\theta$) for state-action pairs not present in the dataset, $\mathcal{D}$:
$$\mathcal{L}_{\text{OOD}} = \mathbb{E}_{s,a\sim p}\left[\hat{r}_\theta(s,a)\right] - \mathbb{E}_{s,a\sim\mathcal{D}}\left[\hat{r}_\theta(s,a)\right]$$

Here, $p$ is a distribution used to sample out-of-distribution state-action pairs. The first term in $\mathcal{L}_{\text{OOD}}$ penalizes high predicted reward values for out-of-distribution pairs, while the second term prevents the learned reward function from collapsing to large negative values. Without the second term, the learned reward function could trivially assign large negative values to all the state-action pairs, including those in the dataset.

## C.3 Online Implementation Details

### C.3.1 Stratified Sampling Hueristic

In the online feedback setting with a limited budget, it is crucial to ask for feedback on the trajectories that maximally increase the information provided to the reward function. To achieve this, we maintain a dataset of the latest 50 trajectories and propose the following trajectory sampling heuristic:

1. **Sorting:** We first sort the trajectories according to their predicted returns.
2. **Sampling:** We then sample $1/3$ of the trajectories from the top 30% of this sorted set, and $2/3$ of the trajectories are sampled from the remaining 70%.
3. **Sub-trajectory selection:** For each sampled trajectory, we extract a sub-trajectory of length $\delta$. This sub-trajectory is either chosen (1) uniformly at random, or (2) as the sub-trajectory with the highest predicted return. Each of these option is applied with equal probability (0.5).

Such a querying mechanism ensures that the queries capture both the typical behavior of the agent and highly informative segments.

### C.3.2 Dynamic Feedback Schedule

We apply a dynamic feedback schedule, collecting feedback more frequently at the beginning of training to provide an initial bias to the reward function. Early feedback helps ground the reward model in the environment and mitigates the impact of random neural network initialization on the agent's learning. Then, as training progresses, we gradually reduce the feedback frequency. This ensures that the reward model is updated with more informative trajectory segments as the RL agent is given more time to adapt to the new reward model after each update.

### C.3.3 ENSEMBLE OF REWARD FUNCTIONS

As is standard practice in many of our baselines (Lee et al., 2021; Park et al., 2022; White et al., 2024), we learn an ensemble of reward functions rather than a single reward function. Each function is trained independently on the same data provided by the simulated teacher. When providing rewards to the agent, we use the mean of the ensemble's outputs.

### C.3.4 REPLAY BUFFER UPDATE

As in previous preference learning methods (White et al., 2024; Lee et al., 2021; Park et al., 2022; Hu et al., 2024), after each reward update, we relabel all samples in the replay buffer with the newly estimated reward. This technique helps reduce the non-stationarity of the RL task and assists the agent's learning process.

### C.4 HYPERPARAMETERS

### C.4.1 SAC HYPERPARAMETERS

**Offline Setting Using SB3 SAC:** We use the default SB3 SAC parameters for the offline experiments.

Table 7: Hyperparameters of SB3 SAC

| Hyperparameter | Value | Hyperparameter | Value |
|---|---|---|---|
| Policy | MLP | Critic target update freq | 1 |
| Init temperature | 0.1 | Critic EMA | 0.005 |
| Learning rate | 3e-4 | Discount | 0.99 |
| Batch size | 256 | | |

**Online Setting Using PEBBLE SAC:** We use the default SAC parameters mentioned in Hu et al. (2024):

Table 8: Hyperparameters of PEBBLE SAC

| Hyperparameter | Value | Hyperparameter | Value |
|---|---|---|---|
| Discount | 0.99 | Critic target update freq | 2 |
| Init temperature | 0.1 | Critic EMA | 0.005 |
| Alpha learning rate | 1e-4 | Actor learning rate | 5e-4 (Walker_walk, Cheetah_run) |
| Critic learning rate | 5e-4 (Walker_walk, Cheetah_run) | | 1e-4 (Other tasks) |
| | 1e-4 (Other tasks) | Actor hidden dim | 1024 |
| | | Actor hidden layers | 2 |
| Critic hidden dim | 1024 | Batch size | 1024 |
| Critic hidden layers | 2 | Optimizer | Adam Kingma & Ba (2015) |

### C.5 REWARD LEARNING HYPERPARAMETERS

**Offline Setting:**

Table 9: Common Hyperparameters

| Hyperparameter | Value | Hyperparameter | Value |
|---|---|---|---|
| $B$ | 64 | Ranking regularization Blondel et al. (2020) | 1.0 |

Table 10: Model Architecture

| Model | #Hidden layers | #Hidden units | Intermediate Activation | Final Activation |
|-------|----------------|---------------|-------------------------|------------------|
| Medium | 1 | 10 | ReLU Agarap (2018) | N/A |
| Large (Offline) | 1 | 100 | ReLU | N/A |
| Large (Online) | 1 | 100 | ReLU | Tanh |

Table 11: Offline Hyperparameters

| Environment | #Reward Updates | Model | #Bins |
|-------------|-----------------|-------|-------|
| Reacher | 15000 | Medium | 4 |
| InvertedDoublePendulum | 3000 | Medium | 4 |
| HalfCheetah | 1000 | Large (Offline) | 6 |

**Online Setting:** As mentioned in the main text, we collect the initial (first 40) feedback in finer bins. Later, we merge the bins to be coarser. Here, we mention the return ranges the simulated teacher uses to assign trajectories into bins:

`Walker-walk`: {start:[0, 10, 20, 30, 40, 50, 60, 80, 100, 150, 200, 300, 400, 500, 600, 800, 1000], end:[0, 30, 60, 100, 200, 300, 400, 500, 600, 800, 1000] }

`Walker-stand`: {start:[0, 100, 130, 140, 150, 160, 170, 200, 300, 400, 500, 600, 800, 1000], end: [0, 100, 140, 160, 200, 300, 400, 500, 600, 800, 1000]}

`Humanoid-stand`: [0, 0.01, 10, 30, 50, 80, 100, 150, 200, 300, 400, 500, 600, 800, 1000]

`Cheetah-run`, `Quadruped-walk/run`: {[0, 5, 10, 20, 30, 40, 50, 60, 80, 100, 150, 200, 300, 400, 500, 600, 800, 1000], end:[0, 30, 60, 100, 200, 300, 400, 500, 600, 800, 1000]}

# D  LLM USE

We used an LLM to edit writing at the sentence level.

Table 12: Online Hyperparameters

| Hyperparameter | Value | Hyperparameter | Value |
|----------------|-------|----------------|-------|
| Training Steps | 2M (Humanoid, Quad-ruped), 1M (Other Tasks) | Reward Updates | 500 (Humanoid), 1000 (Others) |
| | | #Preference per session | 20 (Humanoid), 10 (Other tasks) |
| Reward L2 ($\beta$) | 0.005 (Humanoid), 0.01 (Other tasks) | | |
| | | #Bins | Dynamic |
| $\delta$ | 50 | Reward Learning Rate | 3e-4 |
| Model | Large (Online) | Reward Optimizer | Adam |
| Ensamble size | 3 | | |

