# OpenReview forum: "Reward Learning through Ranking Mean Squared Error"
_ICLR.cc/2026/Conference — Submitted to ICLR 2026_

### Official Review · Reviewer_ypij · 2025-10-26

**Soundness:** 2
**Presentation:** 3
**Contribution:** 2
**Rating:** 4
**Confidence:** 3

**Summary:**

The paper proposes Ranked Return Regression (R4), which trains a reward model from multi-class trajectory ratings using a ranking mean squared error (rMSE) loss built on soft ranks, and claims theoretical guarantees plus empirical gains on locomotion and DMC benchmarks. While this paper solves reward modeling by using the MSE loss of ordinal feedback, it is not a practical method because it requires human rating (classification) of each trajectory.

**Strengths:**

1. The idea of using ranking information from ratings is clear and simple.

2. The method is easy to implement within standard reward learning pipelines.

3. The experiments show some improvement over the chosen baselines on locomotion tasks under simulated settings.

**Weaknesses:**

1. The ordinal feedback is not a new idea for preference-based RL. The paper does not convincingly show that rMSE gives qualitatively different behavior than straightforward regression-to-targets, ordinal regression, or a calibrated cross-entropy approach.

2. There is a lack of related work [1,2]. [1] is an important one that theoretically analyzes why ordinal feedback can work better than pair-wise ones. Meanwhile, some work has explored using tied preference to improve PbRL [2], which should be included in related work and compared with R4.

[1] Liu, S., Pan, Y., Chen, G., & Li, X. Reward Modeling with Ordinal Feedback: Wisdom of the Crowd. 2025

[2] Liu, J., Ge, D., and Zhu, R. Reward learning from preference with ties. 2024

3. The most serious problem is that ordinal feedback is impractical in practice. This paper only improves the loss function without fundamentally solving the problem. Trajectory rating for real humans is a difficult and error-prone task, and solving this problem is the key to ordinal feedback.

4. The experiments use a scripted teacher to provide rates for trajectories, which can demonstrate R4's improvement over RBRL. However, it can not fully convince that R4 is better than comparison-based methods. Although they use the same number of feedback (number of rating trajectories = number of comparisons), this is not a fair comparison, as ratings are much more difficult to compare for a real human. Therefore, it is extremely important to prove the robustness of this method. The paper must show sensitivity to label noise, inter-rater variance, and biased raters (human-in-the-loop experiments); otherwise, claims about “less human effort” or “better sample efficiency” are ungrounded.

5. The experiment results are all curves but lack the IQM result report [3], which has the ability to accurately measure task performance across different tasks and seeds.

[3] M., Castro, P. S., Courville, A. C., & Bellemare, M. Deep reinforcement learning at the edge of the statistical precipice. 2021

6. The paper does not provide pseudocode, which makes it hard to understand.

**Questions:**

1. How sensitive is the method to the soft ranking temperature and other hyperparameters?

2. How does the method behave with strong class imbalance or missing classes?

3. How robust is the method to noisy and inconsistent ratings across different raters?

4. What is the relationship between learned reward sums and true returns? Is it more accurate than comparison-based methods?

---

> ### Author Response · Authors · 2025-11-21
> **Response to Reviewer ypij**
>
> **Weakness 1**
>
> We agree that ordinal feedback is not a new idea in preference-based RL, and we do not claim this (or, if we have made a mistake in our text implying this, please let us know so that we can correct it). Our paper explicitly acknowledges prior work that has used ordinal or rating-based feedback, including the RbRL framework. Our contribution is not the introduction of ordinal feedback itself, but rather the development of a new ranking-based loss function and, consequently, a new rating-based RL method, R4. We also specifically compare our approach against more recent and state-of-the-art baselines, including one rating-based RL method and three preference-based RL methods. These comparisons were conducted across six MuJoCo environments, and the results show that R4 consistently achieves higher performance than these baselines.
>
> **Weakness 2**
>
> The referenced method in [2] addresses reward learning in the context of large language models, whereas our work focuses on learning reward models for continuous-control, simulated robotic environments. We therefore chose to compare against baselines that operate in the continuous-control settings. However, we have included [1] in the related works section as this work is related to learning from ordinal feedback.
>
>
> **Weaknesses 3 and 4 and Q2, 3**
>
> We primarily rely on prior work (RbRL [4]) to show that humans can provide ordinal feedback. As discussed in the general comment, we also included a new discussion showing that humans are able to provide ordinal feedback, supported by results from a five-participant pilot study we conducted. While human raters produce noisy ratings that differ from one another, our method can still learn effective reward functions and produce policies comparable to those obtained with simulated, noise-free ratings.
>
> **Question 4**
>
> To compare the similarity of the learned reward functions with the ground-truth rewards, we report the Trajectory Alignment Coefficient (TAC) [5] in Appendix B.3. TAC measures how similarly two reward functions rank a set of trajectories, where 1 indicates perfect alignment and −1 indicates complete misalignment. For this comparison, we consider the trajectories encountered during training for three environments (Walker-walk, Cheetah-run, Humanoid-stand). We found that R4 produces reward functions that are more aligned with the ground-truth reward in two out of three environments compared to the other rating and preference based methods.
>
> **Weakness 5**
>
> To test for significant differences between algorithms, we use one-sided t-tests in Appendix B.3. We also apply a Bonferroni correction for the online experiments as these experiments involve four algorithmic comparisons per environment. We use a corrected significance threshold of alpha/4 = 0.0125 to control the family-wise error rate. This procedure follows the empirical RL best-practice recommendations outlined in [6], ensuring that our statistical analysis accurately accounts for multiple comparisons.
>
> **Question 1**
>
> We use the default soft-ranking temperature provided in their paper and code, without modification for any of our experiments. We will include a sensitivity analysis in the revised paper and get back to you by November 27th.
>
> **References**
>
> [1] Liu, S., Pan, Y., Chen, G., & Li, X. Reward Modeling with Ordinal Feedback: Wisdom of the Crowd. 2025
>
> [2] Liu, J., Ge, D., and Zhu, R. Reward learning from preference with ties. 2024
>
> [4] White, D., Wu, M., Novoseller, E., Lawhern, V. J., Waytowich, N., & Cao, Y. (2024). Rating-Based Reinforcement Learning. Proceedings of the AAAI Conference on Artificial Intelligence.
>
> [5] Muslimani, C., Johnstonbaugh, K., Chandramouli, S., Booth, S., Knox, W. B., & Taylor, M. E. (2025). Towards Improving Reward Design in RL: A Reward Alignment Metric for RL Practitioners. Reinforcement Learning Conference.
>
> [6] Patterson, A., Neumann, S., White, M., & White, A. (2024). Empirical Design in Reinforcement Learning. Journal of Machine Learning Research.

---

> > ### Author Response · Authors · 2025-11-28
> > **Soft-ranking temperature sensitivity**
> >
> > We have now added an analysis of soft-ranking temperature sensitivity in Figure 15 (Appendix B.2).

---

### Official Review · Reviewer_RiUH · 2025-10-28

**Soundness:** 2
**Presentation:** 3
**Contribution:** 2
**Rating:** 4
**Confidence:** 4

**Summary:**

The paper proposes a rating‑based reward‑learning method that treats human ratings as ordinal supervision. Given trajectory–rating pairs, the proposed method R4 predicts returns with a reward model, applies a differentiable ranking operator to obtain soft ranks, and minimizes a ranking mean squared error (rMSE) between these soft ranks and the (ordered) rating labels. The authors argue this avoids hand‑crafted decision boundaries used in Rating‑based RL (RbRL) and preserves within‑class variance. They provide theory claiming the rMSE objective is “minimal and complete” under assumptions (deterministic realizability, correct binning, and an exact differentiable ranking), and evaluate R4 with simulated raters in Gym and DMC locomotion tasks.

**Strengths:**

- Simple, practical objective. Treating ratings as ordered targets and aligning ranks, instead of absolute scores, gives an intuitive loss that removes the design burden of rating bin boundaries required by RbRL.
- Leverages modern differentiable ranking. Building on fast, differentiable ranking/sorting (permutahedron projections) is sensible and computationally efficient compared to older proxies.
- Broad empirical envs on standard control suites. The paper reproduces common baselines (PEBBLE, SURF, QPA) and reports consistent improvements in several tasks, with ablations on feedback scheduling and sampling.

**Weaknesses:**

- The “minimality and completeness” results crucially assume (i) deterministic reward realizability, (ii) perfect ordinal binning by the teacher, and (iii) exact differentiable ranking that outputs true ranks. In practice, (iii) is not guaranteed—soft‑rank operators are precise for soft projections but do not generally equal hard ranks except in limiting or special cases. The paper cites [1] for “exact computation,” but that exactness refers to the soft operator and its gradients, not equality to hard ranks; the assumption that soft ranks equal true ranks is unrealistic for finite regularization and noisy scores.
- All results derive ratings from the ground‑truth reward, which risks optimistic conclusions and sidesteps the very issues that make reward learning hard (ambiguity, inconsistency, bias). Prior work has highlighted reward misspecification and gaming; without human‑rated studies or at least real‑world noisy labels, it is hard to assess robustness.
- Missing or under‑engaged related works: Rating/ordinal RM beyond RbRL: recent ordinal feedback reward modeling for LLMs extends beyond binary preferences and analyzes conditions for unbiasedness with graded labels—highly relevant to “ratings as ordinals.” [2]. VPIL models vague pairwise during the learning [3]. LiPO/PLPO optimize with ranked lists rather than pairs; while policy‑side, they show the broader move from pairwise to listwise/ordinal signals [4].

## References
[1] Fast differentiable sorting and ranking.

[2] Reward Modeling with Ordinal Feedback: Wisdom of the Crowd.

[3] Imitation Learning from Vague Feedback.

[4] LiPO: Listwise Preference Optimization through Learning-to-Rank.

**Questions:**

Please refer to the weaknesses.

---

> ### Author Response · Authors · 2025-11-21
> **Response to Reviewer RiUH**
>
> **Weakness 1**
>
> On lines 320–323 of the paper, we explain that we provide a relaxed version of Theorem 1 for the setting in which Assumption 4 (the exactness assumption) does not hold (Also see Assumption 5). This relaxation is stated formally in Theorem 2 (lines 335–341). The purpose of Assumption 4 and Theorem 1 is primarily to build intuition and provide a clear proof sketch for the idealized case. The text also states that “the requirement that the ranking operator be differentiable arises solely from the use of a gradient-based optimizer; it is not required by the optimization objective itself. A non-gradient-based optimizer could be used with hard rankings” (lines 293-295).  Finally, we justify Assumption 4 in Appendix A.4, where we show that the soft ranks are equal to the hard  ranks for the ranking operator in [1], most of the time.
>
> **Weakness 2**
>
> We have added a small human-subject pilot study (Appendix B.1) with five participants to show that R4 can perform well even with real, noisy human ratings. Our analysis shows that participants provided labels in a class-imbalanced manner and that ratings were noisy. Trajectories with similar ground-truth returns received different ratings, a phenomenon observed for all participants. See the general comment for a full description. Moreover, in Appendix B.2 (Figure 12), we provide experiments with noisy simulated feedback, demonstrating that R4 maintains strong performance even under high levels of noisy feedback (80% noise).
>
> **Weakness 3**
>
> We specifically focused our related work on reward learning in continuous-control RL settings (e.g., simulated robotics). However, the recent work suggested on ordinal-feedback reward learning—though developed in the LLM context—is still relevant, as our method also uses ordinal feedback. Therefore, we have included these references in the submission.

---

### Official Review · Reviewer_T1Nu · 2025-10-28

**Soundness:** 3
**Presentation:** 3
**Contribution:** 2
**Rating:** 4
**Confidence:** 4

**Summary:**

This paper proposes Ranked Return Regression for RL (R4), a rating-based reward learning method that learns reward functions from ordinal human ratings instead of pairwise preferences or manually designed rewards. R4 introduces a ranking mean squared error (rMSE) loss that aligns predicted returns with teacher-provided ratings through differentiable ranking. The authors provide theoretical guarantees of completeness and minimality, and demonstrate through experiments on OpenAI Gym and DeepMind Control Suite that R4 outperforms existing rating- and preference-based methods.

**Strengths:**

1. The proposed ranking mean squared error (rMSE) loss is conceptually simple yet technically elegant, effectively leveraging ordinal ratings through differentiable ranking.

2. The theoretical analysis provides formal guarantees of completeness and minimality, strengthening the method’s conceptual foundation.

3. The experimental results encompass both offline and online feedback settings across multiple continuous-control benchmarks.

4. The ablation studies are carefully designed, demonstrating the stability of the core method and the incremental benefits of auxiliary design choices such as dynamic feedback scheduling and stratified sampling.

**Weaknesses:**

1. The paper provides limited conceptual and empirical discussion on how rating-based feedback compares to preference-based feedback. While the authors claim that ratings are more informative and cognitively efficient, this assumption is not rigorously analyzed.

2. The theoretical results rely on several strong assumptions, such as deterministic reward realizability, perfectly consistent rating bins, and nearly exact differentiable ranking, which may not hold in realistic human feedback scenarios.
I acknowledge that Assumption 5 attempts to relax the requirement of exact ranking by introducing bounded ranking errors, but it only addresses the approximation error of the differentiable ranking operator rather than the inherent noise or inconsistency in human feedback.

3. All experiments are conducted with simulated feedback derived from environment rewards. As a result, the evaluation does not account for real-world human variability, label noise, or subjective inconsistencies that often arise in practice. Moreover, the experimental environments (DMC and MuJoCo) and the rating-based setting are not well aligned. It would be extremely difficult for human annotators to provide consistent absolute ratings or rankings for these continuous-control trajectories, which differ subtly in motion quality rather than in clear success or failure outcomes. This mismatch raises questions about how well the current results translate to realistic human feedback scenarios.

4. The proposed framework's generalization to true human feedback or more complex domains remains unverified, limiting the empirical validity of its broader claims.

**Questions:**

1. How would the proposed rMSE formulation perform when ratings come from real human annotators, whose feedback may be noisy, inconsistent, or influenced by context?
Given that all experiments use simulated feedback, how might the results change in a genuine human-in-the-loop setting?

2. Does the method remain theoretically sound or empirically stable if the assumption of perfectly ordered or deterministic ratings is relaxed?

3. How sensitive is R4 to the number of rating levels or to non-uniform distributions of ratings (e.g., heavy bias toward mid-level ratings)?

4. Conceptually, the current formulation seems to discretize what could otherwise be treated as continuous feedback. To what extent is R4 genuinely modeling human ordinal judgment, as opposed to learning from a discretized version of continuous return values?

Overall, I find the proposed approach well-motivated and technically sound within the ranking-based setting. The method provides a reasonable and effective formulation for learning from ordinal ratings, and the rMSE loss represents a meaningful contribution in this direction. However, the paper would benefit from a deeper conceptual discussion of how rating-based feedback fundamentally differs from preference-based supervision, both in terms of information structure and practical trade-offs. Moreover, the theoretical and experimental analyses rely on strong and idealized assumptions. A broader reflection on these assumptions, or additional experiments that relax them, especially regarding their implications for real human feedback, would substantially strengthen the paper's overall impact and credibility. While I appreciate the noise-related experiments included in the appendix, these simulations do not fully capture the complexity and inconsistency of real human feedback, and thus leave open questions about the method’s robustness in practical settings.

---

> ### Author Response · Authors · 2025-11-21
> **Response to Reviewer T1Nu**
>
> **Weakness #1**
>
> Please see the general comment for our discussion of ratings versus preferences, including why ratings may be more cognitively efficient for participants.
>
> **Weakness #2 and Q2**
>
> We do not provide a theoretical analysis of our loss function under human-like noise, which would require an in-depth study of bias–variance tradeoffs. This is a great idea, but it is out of the scope of this first paper, and we leave it for future work.
>
> **Weaknesses # 3, 4 and Q1, 3**
>
> We have now added a small human-subject pilot study (Appendix B.1) with five participants (two authors and three non-authors) to demonstrate the effectiveness of R4 with real human ratings. Our analysis shows that participants provided labels in a class-imbalanced manner and that ratings were noisy, with trajectories of similar ground-truth returns receiving different ratings from different participants. Please see the general comment for a full description of the pilot study.
>
> In terms of the sensitivity of R4 on the number of rating levels, in Figure 13, Appendix B.2, we tested 2, 4, and 8 bins and did not find significant differences in the performance of R4. Furthermore, previous human-in-the-loop preference and rating-based methods also use continuous-control environments such as DMC and MetaWorld [1-4]. These precedents, along with our user study, demonstrate that providing ratings in continuous-control environments is not as difficult as the reviewer suggests.
>
> **Question 4**
>
> We weren’t sure how to interpret this question and would really appreciate further clarification. This will help us address the point here and/or in the revision.
>
> **References**
>
> [1] Christiano, P. F., Leike, J., Brown, T. B., Martic, M., Legg, S., & Amodei, D. (2017). Deep reinforcement learning from human preferences. Advances in Neural Information Processing Systems.
>
> [2] Hejna, D. J., III, & Sadigh, D. (2023). Few-shot preference learning for human-in-the-loop RL. Conference on Robot Learning.
>
> [3] White, D., Wu, M., Novoseller, E., Lawhern, V. J., Waytowich, N., & Cao, Y. (2024). Rating-Based Reinforcement Learning. Proceedings of the AAAI Conference on Artificial Intelligence.
>
> [4] Muslimani, C., & Taylor, M. E. (2025). Leveraging sub-optimal data for human-in-the-loop reinforcement learning. International Conference on Learning Representations.

---

### Author Response · Authors · 2025-11-21
**Response for questions that concerned all reviewers.**

We thank the reviewers for their thoughtful comments and questions! In light of these, we have run and analyzed a small ethics-approved human subject study, which we discuss below. We also discuss the relative ease of providing ratings versus preferences.

**Effectiveness of our proposed method with real humans**

We have now added a small human-subjects pilot study (Appendix B.1) with five participants (two authors and three non-authors). In this study, participants provided approximately 100-200 ratings for the OpenAI Gym Reacher task, a continuous state-and-action environment, demonstrating that humans can provide ratings for continuous-control trajectories. Participants could specify the number of bins they wanted for their ratings, ranging from 3 to 10. In Figures 5-10 in Appendix B.1, we observed that even with human-provided ratings, our algorithm R4 could (1) outperform the RbRL baseline and (2) perform comparably to R4 when using synthetic ratings. This suggests that our results obtained with synthetic ratings may generalize to settings with human raters.

Future work includes running more participants on more domains, comparing directly with collecting preferences from human subjects, and performing a qualitative analysis to understand where participants prefer one method over another. However, we hope this small study is sufficient to show the promise of R4 and its impact on data from humans (as well as the extensive experiments on synthetic data).


**Imperfect / Non-uniform Data**

In Figures 6-10 in Appendix B.1, we found that participants often rated trajectories in a class-imbalanced manner, providing more labels for one class than another. Moreover, the distribution of labels per rater also varies. This further supports our claim that R4 is likely to be robust to real-world settings with class imbalance and inter-rater variability. These figures also contain the distribution of undiscounted environment returns associated with each label in the rated dataset. We found substantial overlap in true returns across labels. This means that trajectories with a similar ground truth return were labeled differently. This confirms that the human ratings are imperfect/noisy. These results further support our claim that R4 is robust to rater bias and variance.

Our online experiments with simulated feedback also demonstrate that R4 performs well under non-uniform rating distributions. Rating class imbalance occurs naturally since trajectories are added as they are encountered by the agent, which rarely produces uniform coverage. In addition, R4 remains effective even when some rating bins are missing, such as when the agent fails to generate trajectories corresponding to the highest ratings.

**Humans’ ability to provide ratings versus preferences**

Our work builds on the RbRL paper, which provides a detailed analysis of the claim that ratings are more informative and cognitively efficient than preferences. In their human subject study, participants were asked to provide both ratings and binary preferences over trajectories from OpenAI Gym Mujoco environments (Hopper, Swimmer, and Half-Cheetah). The study also included a qualitative survey assessing factors such as frustration and mental demand. The survey responses suggested that participants found the task more frustrating and mentally demanding when providing preferences compared to ratings. Additionally, participants were able to provide more ratings than preferences within the same fixed time period, further supporting the claim that ratings are more cognitively efficient (see [1] for details). When we asked the authors of this paper if we could use their human subject data, we were unfortunately told that it was only used in real-time and was not recorded. However, validating the claims from this prior work is not the primary focus of our paper; we assume they are true and aim to build better rating-based algorithms.

**References**

[1] White, D., Wu, M., Novoseller, E., Lawhern, V. J., Waytowich, N., & Cao, Y. (2024). Rating-Based Reinforcement Learning. Proceedings of the AAAI Conference on Artificial Intelligence.

---

> ### Author Response · Authors · 2025-11-26
> **Gentle reminder to consider our latest rebuttal**
>
> We thank the reviewers for their time in providing the initial set of reviews.
>
> We would like to gently remind the reviewers of our rebuttal responses, and we appreciate your time and consideration in reviewing them.

---

### Comment · Area_Chair_pqcp · 2025-11-25

Dear Reviewers,
Thank you to those who have already begun interacting with the authors — your timely follow-ups are greatly appreciated and reflect the professionalism and care that uphold our community’s standards.
For reviewers who have not yet responded, I would like to offer a gentle reminder. Authors have invested substantial time and effort into preparing their rebuttals, carefully addressing each concern raised in the reviews. As fellow researchers, we all understand the importance of being heard and having our clarifications considered. Even a brief acknowledgment or follow-up question helps ensure that the evaluation remains fair, thorough, and respectful of everyone’s work.
Your engagement during this phase is essential for maintaining a constructive and high-quality review process. Thank you again for your service and for treating both your fellow reviewers and the authors with the same consideration you would hope to receive.
Best regards,
AC

---

### Author Response · Authors · 2025-12-02
**Summary for AC**

We wanted to summarize the primary concerns across the reviews:

**Issues rooted in prior work rather than in our own contributions.**

Several reviewer comments raised concerns about humans’ ability to provide ordinal feedback. Our work builds on claims from the rating-based RL literature [1], which provided evidence that humans can give ratings in continuous-control environments and find ratings less frustrating to provide than preferences. While we have clarified these assumptions and expanded the discussion, a full re-evaluation of prior work is beyond the scope of this submission.


**The need for evaluation with real human feedback.**

 To directly address this concern, we conducted an human-subjects pilot study after receiving the initial reviews. The new results demonstrate that participants can provide ordinal ratings on continuous-control trajectories, and that R4 performs effectively under realistic human noise, class imbalance, and inter-rater variability.
We hope that these additions resolve the substantive issues raised by the reviewers and strengthen the empirical credibility of our work.
Thank you for your consideration.

[1] White, D., Wu, M., Novoseller, E., Lawhern, V. J., Waytowich, N., & Cao, Y. (2024). Rating-Based Reinforcement Learning. Proceedings of the AAAI Conference on Artificial Intelligence.

---

### Meta-Review · Area_Chair_wZ4o · 2025-12-22

**Summary:**

The paper introduces Ranked Return Regression for RL (R4), a framework designed to learn reward functions from trajectory ratings rather than  hand-shaped rewards or pairwise preferences. The core contribution is a ranking mean squared error (rMSE) loss that uses a differentiable ranking operator to align predicted returns represented as "soft ranks" with the teacher-provided rating labels. The main contribution lies in treating
ratings as ordered targets, thus eliminates the need for the hand-crafted decision used in previous work on Rating-based Reinforcement Learning (RbRL). The authors provide a theoretical and empirical analysis, as well as a limited human-based rating experiment provided in rebuttal.

The main reviewer concerns that informed the recommendation for this paper are:

+ The very strong assumptions upon which the theoretical results of the paper rest. These include deterministic reward realizability, perfect ordinal binning, and exact differentiable ranking. The authors failed to provide a convincing argument regarding the reasonableness of these assumptions, which were voiced as major concerns by two out of the three reviewers.
+ The reasonableness of ordinal human feedback. All three reviewers raised this as a major concern, and although the authors provided a limited, pilot study in rebuttal based on ordinal human feedback, the lack of a significant comparison of ordinal versus preference feedback is a major limitation of the contribution. Moreover, the preference-based baselines considered in the comparative evaluation are all evaluated on robotic manipulation tasks, while in this paper (and in the RbRl paper) only MuJoCo-style continuous control tasks are considered.

**Reviewer Concerns:**

Concerns related to the reasonableness of ordinal human feedback were partially addressed in rebuttal. However this oversight in the original submission and the limited pilot experiment on ordinal human feedback does not adequately address the concerns voiced by all reviewers. There is only one prior work cited justifying the reasonableness of ordinal versus preference human feedback, so the case for ordinal feedback is hardly closed at this point in time.

Concerns regarding the strong theoretical assumptions were not adequately addressed in rebuttal, and the empirical evidence in support of them provided by the authors is limited and unconvincing.

**Reviewer Scores:**

+ **R1 (T1Nu)**: Voiced very strong concerns about both the strong assumptions and the justification for human ordinal (versus preference) feedback. I do not believe this reviewer would have been convinced by the rebuttal arguments or the limited human-based pilot experiment reported in rebuttsl.
+ **R2 (RiUH)**: Also raised questions about the strong assumptions, commented on missing discussion of related work, and also questioned the reasonableness of human ordinal judgments (and the lack of empirical thoroughness in the manuscript to demonstrate it). Again, I find it unlikely the reviewer would have been convinced to significantly change their opinion.
+ **R3 (ypij)**: This reviewer's concerns were mostly limited to questions regarding the ability of humans to provide consistent ordinal judgments, as well as some pointers to related work missing from the discussion. Although the reviewer's concerns regarding statistical significance were addressed by the authors in rebuttal, and this might have softened the initial opinion, I do not believe they would have raised their score above a 6 (given their strong objections to the lack of sufficient justification for ordinal human judgments).

---

### Decision · Program_Chairs · 2026-01-26

Reject